# Three learning stages and accuracy–efficiency tradeoff of restricted Boltzmann machines

Lennart Dabelow [1] ✉ & Masahito Ueda [1,2]

Restricted Boltzmann Machines (RBMs) offer a versatile architecture for unsupervised machine learning that can in principle approximate any target probability distribution with arbitrary accuracy. However, the RBM model is usually not directly accessible due to its computational complexity, and Markov-chain sampling is invoked to analyze the learned probability distribution. For training and eventual applications, it is thus desirable to have a sampler that is both accurate and efficient. We highlight that these two goals generally compete with each other and cannot be achieved simultaneously. More specifically, we identify and quantitatively characterize three regimes of RBM learning: independent learning, where the accuracy improves without losing efficiency; correlation learning, where higher accuracy entails lower efficiency; and degradation, where both accuracy and efficiency no longer improve or even deteriorate. These findings are based on numerical experiments and heuristic arguments.

Restricted Boltzmann Machines (RBMs)[1,2] are a versatile and conceptionally simple unsupervised machine learning model. Besides traditional applications, such as dimensional reduction and pretraining[3–6] and text classification[7], they have become increasingly widespread in the physics community[8,9]. Examples include tomography[10,11] and variational encoding[12–18] of quantum states, time-series forecasting[19], and information-based renormalization group transformations[20,21].

A general goal in unsupervised machine learning is to find the best representation of some unknown *target probability distribution* $p(x)$ within a family of *model distributions* $\hat{p}_\theta(x)$, where $\theta$ denotes the model parameters to be optimized. To this end, the RBM architecture introduces two types of *units*, the *visible* units $x = (x_1, \ldots, x_M) \in \mathcal{X}$, which relate to the states of the target distribution, and the *hidden* units $h = (h_1, \ldots, h_N) \in \mathcal{H}$, which mediate correlations between the visible units (see, e.g., refs. 22–24 for reviews and the top-right corner of Fig. 1 for an illustration). We focus on the most common case where both the visible and the hidden units are binary, i.e., $\mathcal{X} = \{0,1\}^M$ and $\mathcal{H} = \{0,1\}^N$. The RBM model is based on a joint Boltzmann distribution for $x$ and $h$,

$$\hat{p}_\theta(x,h) := Z_\theta^{-1}\, e^{-E_\theta(x,h)}, \qquad (1)$$

where the "energy" $E_\theta(x, h) := -\sum_{i,j} w_{ij} x_i h_j - \sum_i a_i x_i - \sum_j b_j h_j$ takes the form of a classical spin Hamiltonian with "interactions" between visible and hidden units described by the *weights* $w_{ij} \in \mathbb{R}$ and "external fields" for visible and hidden units described by the *biases* $a_i, b_j \in \mathbb{R}$. The weights and biases constitute the model parameters $\theta = (w_{ij}, a_i, b_j)$, and the normalization factor

$$Z_\theta := \sum_{x,h} e^{-E_\theta(x,h)} \qquad (2)$$

is referred to as the *partition function*. The model distribution $\hat{p}_\theta(x)$ that approximates the target $p(x)$ is obtained from marginalization over the hidden units, $\hat{p}_\theta(x) := \sum_h \hat{p}_\theta(x,h)$.

The major drawback of RBMs is that the computational cost to evaluate $Z_\theta$ (and hence $\hat{p}_\theta(x,h)$ and $\hat{p}_\theta(x)$) scales exponentially with

[1]RIKEN Center for Emergent Matter Science (CEMS), Wako, Saitama 351-0198, Japan. [2]Department of Physics and Institute for Physics of Intelligence, Graduate School of Science, The University of Tokyo, Bunkyo-ku, Tokyo 113-0033, Japan. ✉e-mail: lennartjustin.dabelow@riken.jp

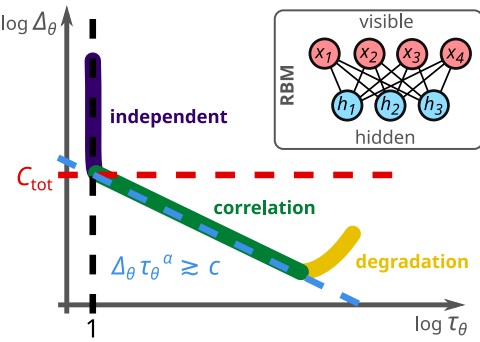

**Fig. 1 | Schematic illustration of the three learning regimes.** These learning stages of Restricted Boltzmann Machines (RBMs) are characterized by the relationship between the model's divergence $\Delta_\theta$ from the target distribution (accuracy, cf. Eq. (3)) and its integrated autocorrelation time $\tau_\theta$ (efficiency, cf. Eq. (6)): independent learning with improved accuracy at no efficiency loss, correlation learing with a power-law tradeoff relation between accuracy and efficiency, and the degragation regime with steady or diminishing accuracy and loss of efficiency. Inset: Schematic illustration of the RBM structure comprised of visible and hidden units.

min$\{M,N\}$ (see also Methods), which renders the model intractable in practice[25]. Therefore, both training (i.e., finding the optimal $\theta$) and deployment (i.e., applying a trained model) rely on approximate sampling from $\hat{p}_\theta(x)$, typically via Markov chains. Ideally, one wishes to generate samples both *efficiently*, in the sense of minimal correlation and computational cost, and *accurately* in the sense of a faithful representation of the target $p(x)$. Unfortunately, these two goals generally compete and cannot be achieved simultaneously.

In this work, we explore the tradeoff relationship between accuracy and efficiency by identifying three distinct regimes of RBM training as illustrated in Fig. 1: (i) independent learning, where the accuracy can be improved without sacrificing efficiency; (ii) correlation learning, where higher accuracy entails lower efficiency, typically in the form of a power-law tradeoff; and (iii) degradation, where limited expressivity, overfitting, and/or approximations in the learning algorithm lead to reduced efficiency with no gain or even loss of accuracy.

Biased or inefficient sampling is a known limitation of standard training algorithms[23,26,27], but it is not an artifact of deficient training methods. Rather, it should be understood as an intrinsic limitation of

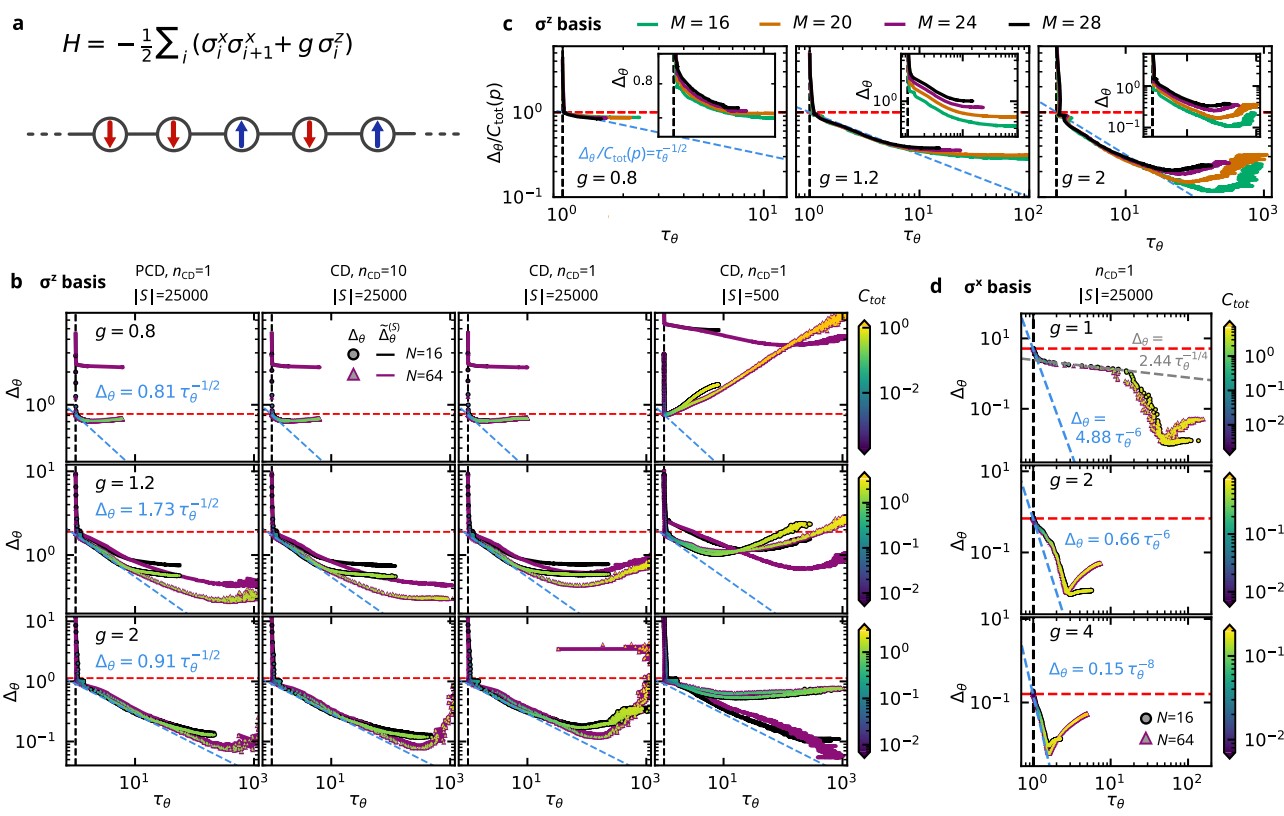

**Fig. 2 | RBM learning characteristics for a quantum-state tomography task.** The ground state of the transverse-field Ising chain with $M$ lattice sites is reconstructed from magnetization measurements along a fixed axis, namely the $z$ direction in **b**, **c** and the $x$ direction in **d**. Thus the ground state is represented in the eigenbases of the $\sigma_i^z$ or $\sigma_i^x$ Pauli operators associated with each lattice site. Training used contrastive divergence (CD) or persistent CD (PCD) with $\eta = 10^{-3}$, $B = 100$. **a** Hamiltonian and sketch of the transverse-field Ising chain, whose ground-state wave function $\psi(x)$ is the square root of the target distribution $p(x)$. **b** Exact loss $\Delta_\theta$ (points) and empirical loss $\tilde{\Delta}_\theta^{(S)}$ (solid lines) vs. autocorrelation time $\tau_\theta$ defined in (6), utilizing PCD (first column) or CD (last three columns), $n_{CD} = 10$ (second column) or $n_{CD} = 1$ (all other columns) and $|S| = 25,000$ (first three columns) or $|S| = 500$ (fourth column) training samples, measured in the $\sigma^z$ basis, for several different values of the magnetic field

$g$ (see left panel of each row). Markers: $\Delta_\theta$ calculated from (3) with the filling color indicating the total correlation $C_{tot}(\hat{p}_\theta)$ of the model distribution (see right colorbars), and the border color and marker type indicating the number of hidden units $N$ (see second panel in first row). Solid lines: $\tilde{\Delta}_\theta^{(S)}$ (see below Eq. (4)), partially masked under the $\Delta_\theta$ data and thus not visible. Dashed lines: $\tau_\theta = 1$ (black), $\Delta_\theta = C_{tot}(p)$ (red), $\Delta_\theta = c \tau_\theta^{-\alpha}$ (blue). **c** $\Delta_\theta/C_{tot}(p)$ vs. $\tau_\theta$ for various system sizes $M$ utilizing CD with $n_{CD} = 1$, $N = 16$, $|S| = 25,000$, $\sigma^z$ basis, and $g$ as indicated in each panel. As a result of rescaling the loss $\Delta_\theta$ with the total correlation $C_{tot}(p)$ of the target distribution, the learning curves collapse in the independent- and correlation-learning regimes. Inset: Same data, but without the rescaling. **d** $\Delta_\theta$ vs. $\tau_\theta$ for CD training in the $\sigma^x$ basis, with $|S| = 25,000$ samples and $n_{CD} = 1$. Markers and dashed lines as in **b**. All curves correspond to averages over 5 independent training runs.

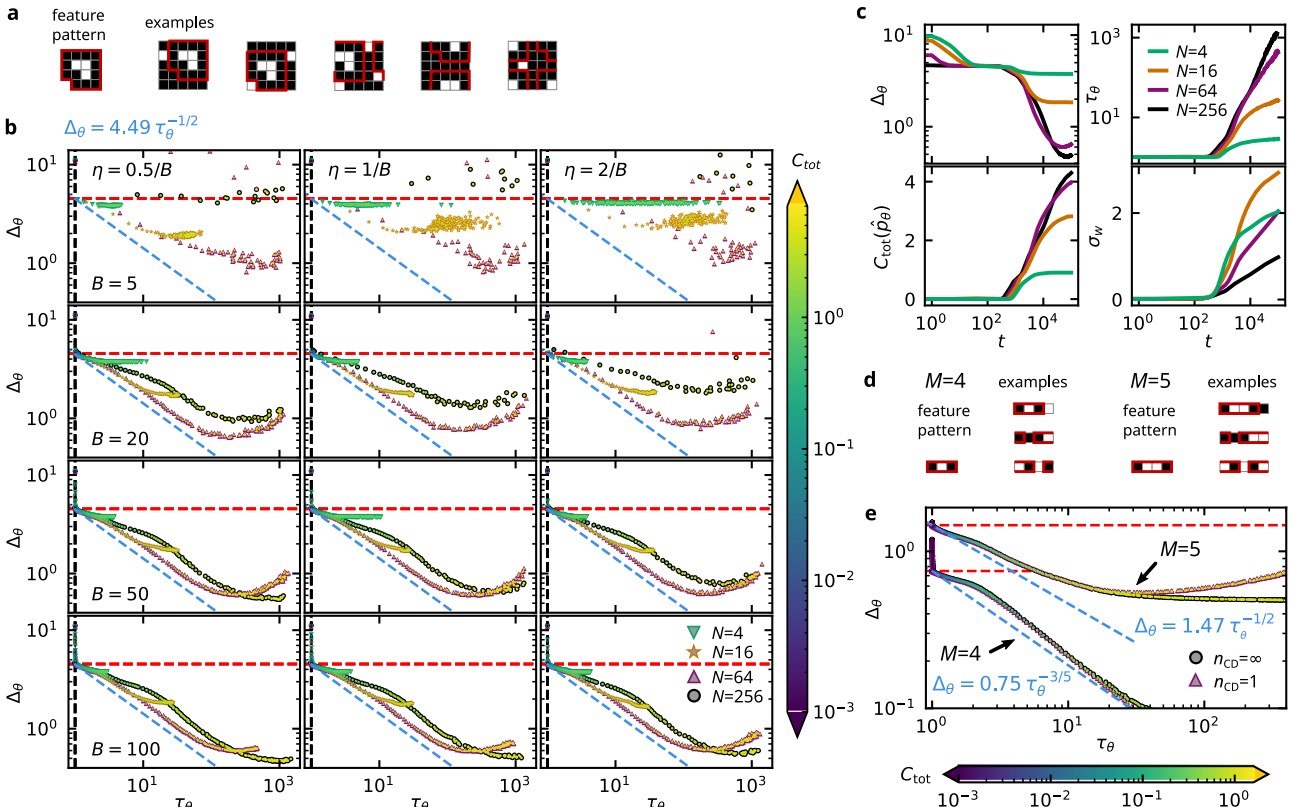

**Fig. 3 | RBM learning characteristics for a pattern recognition task. a** The target distribution consists of $M = 5 \times 5$ "images" subject to periodic boundary conditions and a fixed 15-pixel "hook" pattern implanted at random locations, where the remaining pixels are active (white) with probability $q = 0.1$. **b** Exact loss $\Delta_\theta$ vs. autocorrelation time $\tau_\theta$ for RBMs with different numbers of hidden units $N$ (see legend), trained on the distribution from **a** using contrastive divergence of order $n_{CD} = 1$ with $|S| = 5000$ training samples and various values of the batch size $B$ (rows) and learning rate $\eta$ (columns). Data points are averages over five independent runs. **c** $\Delta_\theta$, $\tau_\theta$, total correlation $C_{tot}(\hat{p}_\theta)$ of the model distribution, and the standard deviation of the weights $\sigma_w := (\frac{1}{MN-1}\sum_{i,j} w_{ij}^2)^{1/2}$ as a function of the training epoch $t$ for various $N$; $\eta = 0.005$, $B = 100$ (cf. bottom left panel of **b**). **d** Simplified model of $M = 1 \times 4$ or $M = 1 \times 5$ images with an implanted "black-white(-white)-black" pattern. **e** $\Delta_\theta$ vs. $\tau_\theta$ for RBMs with $N = 2$ hidden units trained on the distributions from **d** using the full target distribution (i.e., $|S| = \infty$) and exact continuous-time gradient descent with either the full model distribution $\hat{p}_\theta(x,h)$ ($n_{CD} = \infty$) or contrastive divergence of order $n_{CD} = 1$. Data points are averages over 100 independent runs with different initial conditions. **b, e** Fill colors indicate the total correlation $C_{tot}(\hat{p}_\theta)$ of the model distribution (see colorbars), border colors and marker types indicate the number of hidden units $N$ (see legends in bottom-right corners).

the RBM model. Yet its consequences for the usefulness of trained models in applications have received relatively little attention thus far. Our observations (i)–(iii) above elucidate the inner workings of RBMs and imply that, depending on the intended applications, aiming at maximal accuracy may not always be beneficial. We demonstrate the various aspects of these findings by way of several problems, ranging from quantum-state tomography for the transverse-field Ising chain (TFIC, cf. Fig. 2) to pattern recognition and image generation (Figs. 3 and 4); see also the figure captions and Methods for more details on the examples.

## Results

### Accuracy and efficiency

A natural measure for the accuracy of the model distribution $\hat{p}_\theta(x)$ is its Kullback-Leibler divergence $D_{KL}(p\|\hat{p}_\theta)$[28] with respect to the target distribution $p(x)$,

$$\Delta_\theta := D_{KL}(p\|\hat{p}_\theta) \equiv \sum_x p(x) \log \frac{p(x)}{\hat{p}_\theta(x)}, \quad (3)$$

which is nonnegative and vanishes if and only if the distributions $p(x)$ and $\hat{p}_\theta(x)$ agree. Indeed, $\Delta_\theta$ provides the basis of most standard training algorithms for RBMs such as contrastive divergence (CD)[22,29], persistent CD (PCD)[30], fast PCD[31], or parallel tempering[26,32]. Adopting a gradient-descent scheme with $\Delta_\theta$ as the loss function, one would

ideally update the parameters according to

$$\theta_k(t+1) - \theta_k(t) = -\eta \left[ \left\langle \frac{\partial E_\theta(x,h)}{\partial \theta_k} \right\rangle_{\hat{p}_\theta(h|x)p(x)} - \left\langle \frac{\partial E_\theta(x,h)}{\partial \theta_k} \right\rangle_{\hat{p}_\theta(x,h)} \right], \quad (4)$$

where $\eta > 0$ is the learning rate and $\hat{p}_\theta(h|x)$ is the conditional distribution of the hidden units given the visible ones. Since this conditional distribution factorizes and the dependence on $Z_\theta$ cancels out (see Methods for explicit expressions), the first average on the right-hand side of (4) can readily be evaluated. More precisely, since $p(x)$ is unknown, it needs to be approximated by the empirical distribution $\tilde{p}(x; S) := \frac{1}{|S|}\sum_{\tilde{x}\in S} \delta_{x,\tilde{x}}$ for a (multi)set of *training data* $S := \{\tilde{x}^{(1)}, \dots \tilde{x}^{(|S|)}\}$, which are assumed to be independent samples drawn from $p(x)$. Hence the effective loss function is $\tilde{\Delta}_\theta^{(S)} := \sum_x \tilde{p}(x; S) \log \frac{\tilde{p}(x;S)}{\hat{p}_\theta(x)}$, which is an empirical counterpart of (3).

The second average in (4), however, requires the full model distribution (1) and is thus not directly accessible in practice. Instead, it is usually approximated by sampling alternately from the accessible conditional distributions $\hat{p}_\theta(h|x)$ and $\hat{p}_\theta(x|h)$, leading to a Markov chain of the form

$$x^{(0)} \to h^{(0)} \to x^{(1)} \to h^{(1)} \to \cdots \quad (5)$$

The distribution of $(x^{(n)}, h^{(n)})$ converges to the model distribution $\hat{p}_\theta(x,h)$ as $n \to \infty$. Truncating the chain (5) at a finite $n = n_{CD}$, we obtain a

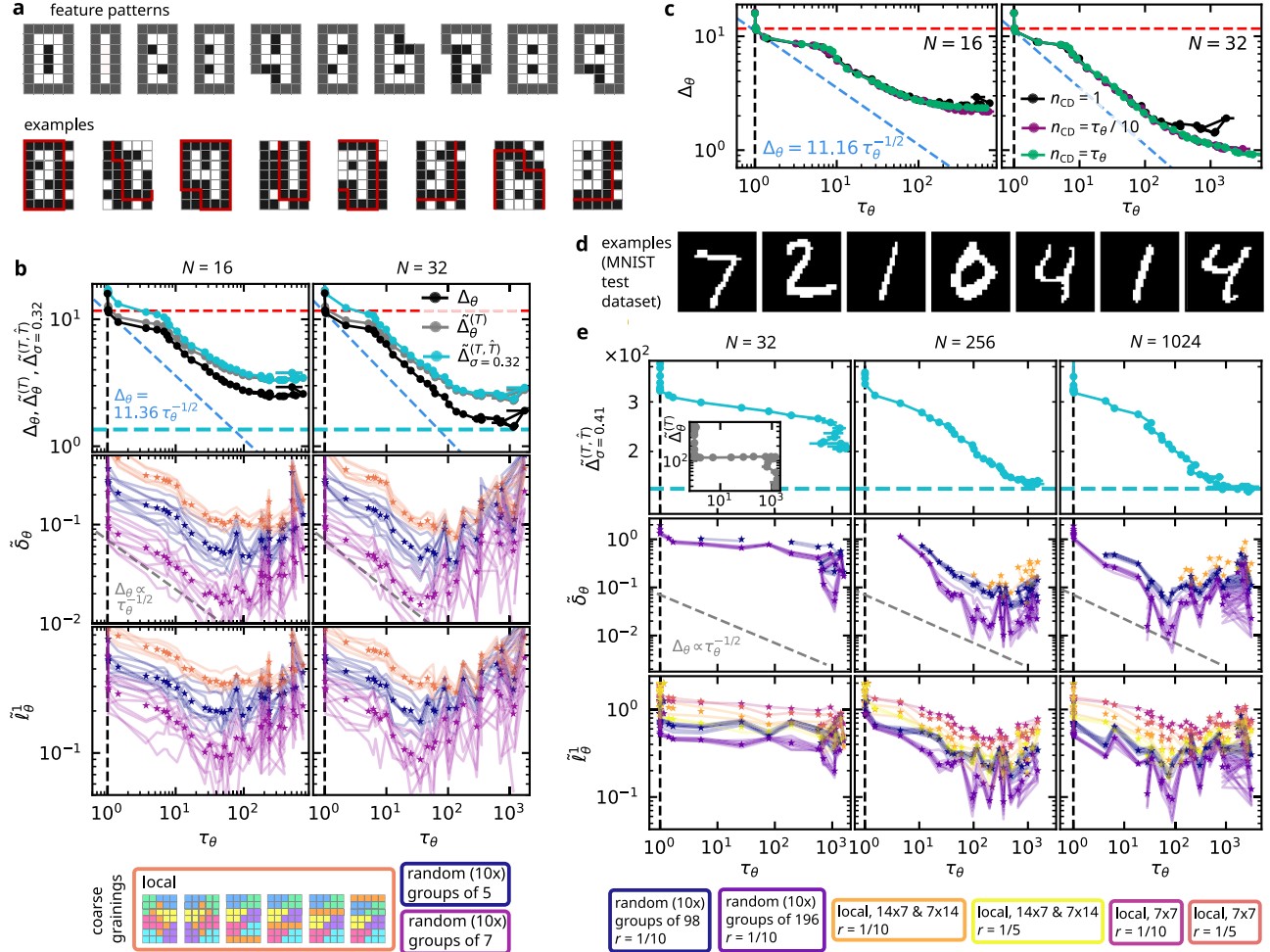

**Fig. 4 | Approximate RBM learning characteristics on digit images. a** Images of $M = 5 \times 7$ pixels showing patterns of the digits 0 through 9 (selected uniformly) at a random location. Gray pixels must either be made black ($x_i = 0$) or be cut away by the image boundaries (see examples in the second row). Pixels that are not part of the pattern are active (white) with probability $q = 0.1$. The total number of such images is 40,507,353. **b** Various loss measures vs. autocorrelation time $\tau_\theta$ for $N = 16$ (left) and $N = 32$ (right) hidden units, utilizing persistent contrastive divergence (PCD) with $n_{CD} = 1$, $\eta = 0.005$, $B = 100$ on $|S| = 50,000$ training images. Top: Exact loss $\Delta_\theta$ (black), exact test error $\tilde{\Delta}_\theta^{(T)}$ (empirical loss for a test dataset $T$ of $|T| = 10,000$ images, gray), and Gaussian-smoothened empirical loss estimate $\tilde{\Delta}_\sigma^{(T,\hat{T})}$ ($|\hat{T}| = 10^6$, $\sigma = 0.32$, cyan). The cyan dashed line marks $\tilde{\Delta}_{\sigma=0.32}^{(T,S)} = 1.354$, the minimal Gaussian-smoothened loss estimate between the test and training datasets. Middle: empirical

error $\tilde{\delta}_\theta$ using majority-rule ($r = 1$) coarse-grainings of samples from the target and model distributions, partitioning pixels into local or random groups (see main text for details). Solid lines: results for individual partitions; star markers: average of the solid lines of the same partitioning type (color, see legend). Bottom: empirical error $\tilde{\ell}_\theta^1$ for the same coarse-grainings. **c** $\Delta_\theta$ vs. $\tau_\theta$ for $N = 16$ (left) and $N = 32$ (right) hidden units, utilizing persistent contrastive divergence (PCD) with fixed ($n_{CD} = 1$) or adaptive ($n_{CD} \propto \tau_\theta$) approximation order; other hyperparameters as in **b**. **d** Examples from the MNIST dataset, which comprises images of $M = 28 \times 28$ pixels showing handwritten digits. **e** Similar to **b**, but for the MNIST dataset and PCD training with $n_{CD} = 1$, $\eta = 10^{-4}$, $B = 100$, $|S| = 60,000$, $|T| = 10,000$, $|\hat{T}| = 10^6$, $\sigma = 0.41$, and $\tilde{\Delta}_{\sigma=0.41}^{(T,S)} = 147.4$. Missing data points correspond to $\tilde{\delta}_\theta = \infty$ and/or unreliable $\tau_\theta$ estimates.

(biased) sample from that distribution, whose bias vanishes as $n_{CD} \to \infty$[33], but depends on the initialization of the chain for finite $n_{CD}$. In our numerical examples, we will usually adopt the common CD algorithm, which chooses $x^{(0)}$ as a sample from the training data $S$, or the PCD algorithm, where $x^{(0)}$ is a sample from the chain of the previous update step (see also Supplementary Note 1). Subsequently, the Markov chain (5) can be used to generate a new, but correlated sample. Similarly, when analyzing and deploying a model $\hat{p}_\theta(x)$ after training, new samples are typically generated by means of Markov chains (5), with the caveat that those samples are correlated and thus not independent.

To quantify the *sampling efficiency*, we therefore consider the integrated autocorrelation time[34]

$$\tau_\theta := 1 + 2 \sum_{n=1}^{\infty} \frac{g_\theta(n)}{g_\theta(0)}, \qquad (6)$$

where $g_\theta(n) := \frac{1}{M} \sum_i [\langle x_i^{(0)} x_i^{(n)} \rangle - \langle x_i^{(0)} \rangle^2]$ is the mean correlation function of the visible units for the Markov chain (5) in the stationary regime, i.e., $x^{(0)} \sim \hat{p}_\theta(x)$. Notably, $\tau_\theta$ is independent of the training algorithm since it depends only on the RBM parameters $\theta$, but not on the different initialization schemes of the Markov chains in (P)CD and its variants. In practice, particularly when utilizing the scheme (5) to employ a trained model productively, one will start from an arbitrary distribution and discard a number of initial samples (ideally on the order of the mixing time[33,35,36]) to thermalize the chain and approach the stationary distribution $\hat{p}_\theta(x)$.

The interpretation of $\tau_\theta$ as a measure of sampling efficiency is as follows: Suppose we have a number $R$ of *independent* samples from the model distribution $\hat{p}_\theta(x)$ to estimate $\langle x_i \rangle$ (or $\frac{1}{M} \sum_i \langle x_i \rangle$). To obtain an estimate of the same quality via Gibbs sampling according to (5), we would then need on the order of $\tau_\theta R$ correlated Markov-chain samples (see, for example, Sec. 2 of ref. 34 and also Methods). Hence the (minimal) value of $\tau_\theta = 1$ hints at independent (uncorrelated) samples,

and the larger $\tau_\theta$ becomes, the more samples are needed in principle, rendering the approach less efficient.

Note that the integrated autocorrelation time $\tau_\theta$ defined in Eq. (6) is conceptually related to, but different from the mixing time of the Markov chain (see also Discussion below). Furthermore, different observables (i.e., functions of the visible units $x_i$) generally exhibit different autocorrelation times. As explained in detail in Methods, the quantity $\tau_\theta$ from (6) is a weighted average of the autocorrelation times associated with the observables' elementary variables, namely the individual $x_i$. Hence we expect $\tau_\theta$ to capture the relevant correlations and thus the sampling efficiency in the generic case. The evaluation of other correlation measures introduced below will reinforce this notion. In addition, a quantitative comparison of autocorrelation times for different observables is provided in Supplementary Note 4 for the examples from Figs. 2 and 4a–c.

Our principal object of study is the mutual dependence of $\Delta_\theta$ and $\tau_\theta$ on the parameters $\theta$ for a given target distribution $p(x)$. As outlined above and illustrated in Fig. 1, there are three regimes the machine undergoes during the learning process. Globally, the overall tradeoff between accuracy and efficiency is numerically found to be bounded by a power law of the form

$$\Delta_\theta \tau_\theta^\alpha \gtrsim c, \tag{7}$$

where both $c$ and the exponent $\alpha$ are positive constants whose meaning will be clarified in the following. Moreover, in the correlation-learning regime, $\Delta_\theta$ and $\tau_\theta$ are often directly related by a power law $\Delta_\theta \tau_\theta^{\alpha'} \simeq c'$, where the constants $c'$ and $\alpha'$ are close to $c$ and $\alpha$, respectively.

## Mechanism behind the learning stages

With no specific knowledge about the target distribution, it is natural to initialize the RBM parameters $\theta = (\theta_k) = (w_{ij}, a_i, b_j)$ randomly. Moreover, the initial values should be sufficiently small so that any spurious correlations arising from the initialization are much smaller than the actual correlations in the target distribution and can be overcome within a few training steps. In the examples from Figs. 2–4, we draw the initial $\theta_k$ independently from a normal distribution $\mathcal{N}(\mu, \sigma)$ of mean $\mu$ and standard deviation $\sigma$, namely $w_{ij} \sim \mathcal{N}(0, 10^{-2})$ and $a_i, b_j \sim \mathcal{N}(0, 10^{-1})$ unless stated otherwise. A brief exploration of other initialization schemes, including Hinton's proposal[22] and examples with significant (spurious) correlations, can be found in Supplementary Note 3. In Figs. 2 and 3, the experiments were repeated for 5 independent runs for each hyperparameter configuration, and the displayed data are averages over those runs at fixed training epoch $t$. No error bars are shown in these figures for clarity, but the spread of the point clouds typically serves as a decent visualization of the uncertainty. We also highlight that important information for the ensuing discussion is encoded in the coloring of the data points. Particularly, both the filling color and the border color convey correlation characteristics and hyperparameter dependencies as indicated in the legends and figure captions.

We now sketch how the three learning regimes and the tradeoff relation arise. Intuitively, the origin of the accuracy–efficiency tradeoff can be understood as follows: To improve the model representation $\hat{p}_\theta(x)$ of the target distribution $p(x)$, correlations of $p(x)$ between the different visible units $x_i$ have to be incorporated into $\hat{p}_\theta(x)$. Since correlations between visible units are mediated by the hidden units in the RBM model (1), this inevitably increases the correlation between subsequent samples in the Markov chain (5) and thus leads to larger autocorrelation times $\tau_\theta$ in (6). Nevertheless, the detailed relationship between $\Delta_\theta$ and $\tau_\theta$ and its remarkable structural universality turn out to be more subtle as discussed in the following.

In the *independent-learning regime*, which constitutes the first stage of the natural learning dynamics, the loss $\Delta_\theta$ is actually reduced without any significant increase of the autocorrelation time $\tau_\theta$. Hence

the RBM picks up aspects of the target distribution whilst preserving independence of its visible units. The minimal loss $\Delta_\theta$ that can be achieved with a product distribution of independent units $x_i$ is given by the *total correlation*[37]

$$C_{\text{tot}}(p) := \sum_x p(x) \log \frac{p(x)}{p_1(x_1) \cdots p_M(x_M)} \tag{8}$$

of the target distribution. This quantity is thus the KL divergence (cf. Eq. (3)) from the product of marginal distributions $p_i(x_i)$ to the joint distribution $p(x) = p(x_1, \ldots, x_M)$. It can be understood as a multivariate analog of mutual information. For an arbitrary product distribution $\hat{p}(x) := \prod_i \hat{p}_i(x_i)$, we have $D_{\text{KL}}(p \| \hat{p}) = C_{\text{tot}}(p) + \sum_i D_{\text{KL}}(p_i \| \hat{p}_i) \geq C_{\text{tot}}(p)$ (see Supplementary Note 5). Hence $C_{\text{tot}}(p)$ indeed lower-bounds the loss $\Delta_\theta$ for independent units.

The value of $C_{\text{tot}}(p)$ is indicated by the red dashed lines in Figs. 1–4, and indeed marks the end of the independent-learning regime as defined by $\tau_\theta \simeq 1$ in Figs. 2–4. As a consequence, we can identify the constant $c$ from the tradeoff relation (7), which bounds $\Delta_\theta$ from below at $\tau_\theta = 1$ (see also Methods), with the total correlation $C_{\text{tot}}(p)$ of the target distribution, $c \simeq C_{\text{tot}}(p)$, as illustrated by the intersection of the red ($\Delta_\theta = C_{\text{tot}}(p)$), blue ($\Delta_\theta = c \tau_\theta^{-\alpha}$), and black ($\tau_\theta = 1$) dashed lines in Figs. 1–4.

A closer inspection of the total correlation $C_{\text{tot}}(\hat{p}_\theta)$ of the *model* distribution, encoded by the color gradients in Figs. 2 and 3, confirms that no significant correlations between the RBM's visible units build up as long as $\tau_\theta \simeq 1$, providing further justification for labeling this stage as the "independent-learning" regime. The time spent in this regime can be reduced by adjusting the biases $a_i$ to the activation frequencies of the visible units in the training data as suggested by Hinton[22] (see also Supplementary Note 3).

The independent-learning regime is thus characterized by $\tau_\theta \simeq 1$ and $\Delta_\theta \gtrsim C_{\text{tot}}(p)$. As soon as $\Delta_\theta$ falls below $C_{\text{tot}}(p)$, the RBM enters the *correlation-learning regime* and starts to exhibit noticeable dependencies between its visible units, accompanied by an increase of $\tau_\theta$. This regime is characterized by $\Delta_\theta \lesssim C_{\text{tot}}(p)$ and $\frac{\partial \tau_\theta}{\partial \Delta_\theta} < 0$, meaning that $\tau_\theta$ grows as $\Delta_\theta$ decreases. Quantitatively (cf. Figs. 2b–d, 3b, e, 4b, c), we find that the functional dependence between $\Delta_\theta$ and $\tau_\theta$ is (piecewise) power-law-like and often closely follows the lower bound provided by the tradeoff relation (7).

In most of our examples, the exponent $\alpha$ turns out to be well approximated by $\alpha \simeq \frac{1}{2}$. The notable exception is the example in Fig. 2d of TFIC ground-state tomography in the $\sigma^x$ basis (but not the $\sigma^z$ basis; see figure caption for details), where a value of $\alpha \approx 6 \ldots 8$ seems more appropriate. Roughly speaking, $\alpha$ quantifies how efficiently the prevailing correlations in the target distribution $p(x)$ can be encoded in the RBM model $\hat{p}_\theta(x)$. A larger value of $\alpha$ implies that the tradeoff (7) is less severe, indicating a closer structural similarity of $p(x)$ to the model family $\hat{p}_\theta(x)$.

The relationship between accuracy and efficiency in the correlation-learning regime turns out to be remarkably stable against variations of the architecture or the training details, suggesting that it is indeed an intrinsic limitation of the RBM model whose qualitative details are essentially determined by the target distribution. First, as long as training is stable, the $\Delta_\theta$–$\tau_\theta$ learning trajectories are almost independent of further hyperparameters such as the number of training samples $|S|$, the minibatch size $B$, or the learning rate $\eta$. This is illustrated in Fig. 3b (see Supplementary Note 5 for further examples), which also visualizes how training becomes unstable if $\eta$ or $B$ become too small, leading to underperforming machines with $(\tau_\theta, \Delta_\theta)$ further away from the global bound (7). Second, changing the approximation of the model averages in (4) does not affect the relation between $\Delta_\theta$ and $\tau_\theta$. In fact, approximation schemes that achieve a smaller loss $\Delta_\theta$ increase the autocorrelation time $\tau_\theta$ in accordance with the tradeoff (7). This is exemplified by variations in the order ($n_{\text{CD}}$) and the

initialization (CD vs. PCD) of the training chains (5) in Fig. 2b. Third, as long as the loss is sufficiently above the expressivity threshold (see below), the relationship between $\Delta_\theta$ and $\tau_\theta$ is largely insensitive to the number of hidden units $N$ (see Figs. 2b, d, 3b and 4b, c). Fourth, the learning characteristics appear to be intrinsic to the problem *type*, but not its *size* if a natural scaling for the number of visible units $M$ exists. To this end, we consider the TFIC example and vary the number of lattice sites $M$ in Fig. 2c. While this changes the total correlation $C_{\text{tot}}(p)$ of the target distribution, the rescaled curves of $\Delta_\theta/C_{\text{tot}}(p)$ vs. $\tau_\theta$ collapse almost perfectly onto a single universal curve in the independent- and correlation-learning regimes.

The end of the correlation-learning regime and the crossover into the *degradation regime* is influenced by various (hyper)parameters. An absolute limit for the minimal value of $\Delta_\theta$ results from the class of distributions that can be represented by the RBM. This "expressivity" is controlled by the number of hidden units $N$. For sufficiently large $N$, the RBM model can approximate any target distribution with arbitrary accuracy[24,38–40]; hence there is no absolute minimum for $\Delta_\theta$ in principle. In practice, however, the number of hidden units is limited by the available computational resources. Note that the scaling of this expressivity threshold is analyzed in some detail in ref. 41 for the TFIC example (cf. Fig. 2).

Ceasing accuracy improvement due to limited expressivity is exemplified by Fig. 3b in the stable regime ($B \gtrsim 50$), where we note that the achievable minimal loss decreases significantly from $N=4$ to 16 to 64 (the same behavior can also be observed in Fig. 4b, c). Employing even more hidden units, however, does not facilitate any significant gain in accuracy, and the learning characteristics for $N=256$ in Fig. 3b actually signal slightly worse performance in terms of the accuracy–efficiency tradeoff, i.e., a larger offset from the global lower bound (blue dashed line).

If $N$ is sufficiently large, the approximations leading to a bias of the (exact) update step (4) will usually take over eventually and lead into the degradation regime even if the expressivity threshold has not yet been reached.

The first of those approximations is the use of the empirical distribution $\tilde{p}(x;S)$ in lieu of the unknown true target distribution $p(x)$. This may result in overfitting, a phenomenon common to many machine-learning approaches: The RBM may pick up finite-size artifacts of $\tilde{p}(x;S)$, particularly when the resolution of genuine features in the model distribution approaches the resolution of those features in the empirical distribution. Overfitting is the primary reason for degradation in the fourth column of Fig. 2b, where the size of the training dataset $|S|$ is rather small. Comparing the training error $\tilde{\Delta}_\theta^{(S)}$ (solid lines, see below (4)) with the test error $\Delta_\theta$ (data points, see Eq. (3)), we observe that the former continues to decrease even though the latter actually increases.

In the first three columns of Fig. 2b, by contrast, $\tilde{\Delta}_\theta^{(S)}$ usually follows $\Delta_\theta$ closely (thus the solid lines are often hidden behind the data points). Here, degradation is due to the second limiting approximation of the update step (4), namely the replacement of averages over the model distribution $\hat{p}_\theta(x,h)$ by Markov-chain samples (5). In fact, this is directly related to the definition of $\tau_\theta$ because larger values imply that the chain (5) needs to be run for a longer time in order to obtain an effectively independent sample (see below Eq. (6)). Indeed, smaller losses can be achieved for larger $n_{\text{CD}}$ (second vs. third column). Similarly, at fixed $n_{\text{CD}}$, PCD can reach higher accuracies than CD (first vs. third column; see also Supplementary Note 5).

Finally, we turn to the smallest example from Fig. 3d, e. In this case, we can directly integrate the continuous-time ($\eta=0$) update equations (4) with the full target distribution $p(x)$ (i.e., $|S|=\infty$) and the exact model distribution $\hat{p}_\theta(x,h)$ (i.e., $n_{\text{CD}}=\infty$) for RBMs with $N=2$ hidden units (see also Supplementary Note 1). We again observe a power-law tradeoff between $\Delta_\theta$ and $\tau_\theta$ with $\alpha \simeq \frac{1}{3}\ldots\frac{2}{5}$, limited by the machine's expressivity in the $M=5$, but not in the $M=4$ case. Moreover,

by averaging over $\hat{p}_\theta^{(1)}(x,h) := \hat{p}_\theta(h|x)\sum_{x',h}\hat{p}_\theta(x|h')\hat{p}_\theta(h'|x')p(x')$ instead of $\hat{p}_\theta(x,h)$ in (4), we can adopt the exact CD update of order $n_{\text{CD}}=1$. This reintroduces the correlation bias into the updates and indeed leads to stronger deviations from the power-law behavior for $M=5$, with increasing $\Delta_\theta$ in the degradation regime.

## Towards applications

All examples discussed so far (Figs. 2, 3 and 4a–c) involved only a small number of visible units $M$ so that the accuracy measure $\Delta_\theta$ could be evaluated numerically exactly. In practice, this is impossible because neither the target distribution $p(x)$ nor the model distribution $\hat{p}_\theta(x)$ are directly accessible. In the following, we will sketch how learning characteristics and the accuracy–efficiency tradeoff can be analyzed approximately in applications and apply the ideas, in particular, to the MNIST dataset[42] as a standard machine-learning benchmark of larger problem size (see Fig. 4d, e).

To approximate the accuracy measure $\Delta_\theta$, the target distribution $p(x)$ is usually replaced by the empirical distribution $\tilde{p}(x;T)$ for a (multi)set of test samples $T$ (independent of the training samples $S$). If both $M$ and $N$ become large, $\hat{p}_\theta(x)$ must be approximated by an empirical counterpart as well. To this end, a collection of independent samples from $\hat{p}_\theta(x)$ is needed. Typically, it will be generated approximately by Markov chains (5), which directly leads back to the autocorrelation time $\tau_\theta$ from (6) as a measure for the number of steps required in (5) to obtain an effectively independent sample.

Estimating $\tau_\theta$, in turn, should remain feasible along the lines outlined in Methods even if $M$ and $N$ are large. To be precise, if it turns out to be impossible in practice to reliably estimate $\tau_\theta$, then any conclusions about the model distribution $\hat{p}_\theta(x)$ drawn from Markov chains like (5) are equally unreliable. In other words, if $\tau_\theta$ (or, more generally, the integrated autocorrelation time of the observable of interest) cannot be computed, the trained model itself becomes useless as a statistical model of the target distribution. A particular challenge are metastabilities where the chains spend large amounts of time in a local regime of the configuration space and only rarely transition between those regimes. These can be caused, for instance, by a multimodal structure of the target distribution. If undetected, those metastabilities can lead to vastly underestimated autocorrelation times.

Once a set of (approximately) independent samples $\hat{T}$ from $\hat{p}_\theta(x)$ is available, the KL divergence $D_{\text{KL}}(\tilde{p}(\,\cdot\,;T)||\tilde{p}(\,\cdot\,;\hat{T}))$ can serve as a proxy for $\Delta_\theta$ in principle. In practice, however, this approach will not be viable because this proxy diverges whenever there is a sample $\tilde{x}$ in $T$ which is not found in $\hat{T}$, meaning that the sample size required for $\hat{T}$ will often be out of reach.

We suggest two alternative approaches to mitigate this problem. First, we consider smoothening the empirical model distribution $\tilde{p}(x;\hat{T})$ by convolving it with a Gaussian kernel $k(x;\mu,\sigma) := N_\sigma^{-1}e^{-(x-\mu)^2/2\sigma^2}$, where $N_\sigma := \sum_{d=0}^{M}\binom{M}{d}e^{-d/2\sigma^2}$, leading to $\tilde{p}_\sigma(x,\hat{T}) := \frac{1}{|\hat{T}|}\sum_{\hat{x}\in\hat{T}}k(x;\hat{x},\sigma)$. The KL divergence $\tilde{\Delta}_\sigma^{(T,\hat{T})} := D_{\text{KL}}(\tilde{p}(\,\cdot\,;T)||\tilde{p}_\sigma(\,\cdot\,;\hat{T}))$ then approximates $\Delta_\theta$, where $\sigma$ is chosen so as to make $\tilde{\Delta}_\sigma^{(T,S)}$ minimal, i.e., when using the training data $S$ as the empirical model distribution[26] (see also Supplementary Note 2). As shown in the first row of Fig. 4b, $\Delta_\sigma^{(T,\hat{T})}$ reproduces essentially the same behavior as $\Delta_\theta$ and $\tilde{\Delta}_\theta^{(T)}$.

Second, we propose coarse-graining the samples in $T$ and $\hat{T}$, such that every $\tilde{x}=(\tilde{x}_1,\ldots,\tilde{x}_M)\in T,\hat{T}$ is mapped to a new configuration $\tilde{y}=(\tilde{y}_1,\ldots,\tilde{y}_L)$ with $\tilde{y}_l\in\{0,1\}$ and $L<M$. Denoting the resulting multisets of reduced configurations by $T'$ and $\hat{T}'$, we then consider the KL divergence $\tilde{\delta}_\theta := D_{\text{KL}}(\tilde{p}(\,\cdot\,;T')||\tilde{p}(\,\cdot\,;\hat{T}'))$ of the associated empirical distributions as a qualitative approximation of $\Delta_\theta$. To be specific, in Fig. 4, we employ a weighted majority rule for coarse graining using random or local partitions of the visible units into $L$ subsets, such that $\tilde{y}_l=1$ if a fraction of $r$ or more units in the $l$th subset is active (see Methods for details).

While some of the quantitative details are inevitably lost as a result of the coarse graining, the results in Fig. 4b show that the accuracy measure $\tilde{\delta}_\theta$ still conveys similar learning characteristics as the exact loss $\Delta_\theta$. Remarkably, even the same exponent $\alpha \simeq \frac{1}{2}$ is found to describe the tradeoff between $\tilde{\delta}_\theta$ and $\tau_\theta$ in the correlation-learning regime. On the other hand, the coarse-grained loss $\tilde{\delta}_\theta$ appears to deteriorate somewhat prematurely, especially for the random coarse grainings, indicating that late improvements of $\Delta_\theta$ involve finer, presumably local correlations that cannot be captured by $\tilde{\delta}_\theta$ in these cases.

Furthermore, we also consider the $L^1$ distance $\tilde{\ell}_\theta^1 := \sum_x |\tilde{p}(x; T') - \tilde{p}(x; \hat{T}')|$ between the reduced empirical distributions as an accuracy measure. Its advantage is that—unlike $\tilde{\delta}_\theta$—it does not suffer from divergences when $T' \not\subseteq \tilde{T}'$ (cf. Fig. 4e in particular). As shown in Fig. 4b and e, the $\tilde{\ell}_\theta^1$–$\tau_\theta$ curves qualitatively agree with their $\tilde{\delta}_\theta$–$\tau_\theta$ counterparts and can thus serve as a more stable way to monitor the tradeoff in case of smaller sample sizes.

Inspecting the learning characteristics in the MNIST example from Fig. 4e, we observe that the relationship between the accuracy measures $\tilde{\Delta}_\sigma^{(T,T)}$, $\tilde{\delta}_\theta$, $\tilde{\ell}_\theta^1$ and the efficiency measure $\tau_\theta$ are qualitatively similar as in the simpler example in Fig. 4b, especially for the more expressive RBMs with $N \geq 256$. Notably, we find an initial regime with decreasing $\tilde{\Delta}_\sigma^{(T,T)}$ and $\tilde{\ell}_\theta^1$ at $\tau_\theta = 1$ ($\tilde{\delta}_\theta = \infty$ here due to the aforementioned undersampling problem), followed by an approximately power-law-like tradeoff between accuracy and efficiency, and finally ceasing improvement ($\tilde{\Delta}_\sigma^{(T,T)}$) or deterioration ($\tilde{\delta}_\theta$, $\tilde{\ell}_\theta^1$) at increasing $\tau_\theta$. For $N = 32$, by contrast, the RBM accuracy does not improve much beyond the independent-learning threshold, except for somewhat unstable fluctuations at very late training stages. Hence we expect that the same tradeoff mechanism identified in the small-scale examples from Figs. 2 through 4a–c also governs the behavior of more realistic, large-scale learning problems.

Altogether, our present results suggest a couple of approaches to monitor the accuracy and efficiency in applications with large input dimension $M$. First, we propose estimating the autocorrelation time $\tau_\theta$ at selected epochs during training and stop when it exceeds the threshold set by the available evaluation resources in the intended use case. Second, it may be helpful to train RBMs with smaller numbers of hidden units $N$ so that the test error $\tilde{\Delta}_\theta^{(T)}$ can be evaluated exactly (see also Methods), even though those small-$N$ machines will typically not reach the desired accuracies. Since the onset of the correlation-learning regime and the subsequent initial progression are essentially independent of $N$, the relationship between $\tilde{\Delta}_\theta^{(T)}$ and $\tau_\theta$ for small $N$ can provide an intuition and perhaps even a cautious extrapolation of the behavior for larger $N$. Third, empirical accuracy measures such as $\tilde{\Delta}^{(T,T)}$, $\tilde{\delta}_\theta$ and $\tilde{\ell}_\theta^1$ can assure that the machine is still learning and possibly even map out the beginning of the degradation regime. Fourth, estimates of $\tau_\theta$ can be naturally obtained *en passant* when using the PCD algorithm. These estimates can then be employed to adapt the length $n_{CD}$ of the Markov chains (5) to the current level of correlations when approximating the model averages in (4). While we leave a detailed analysis of the resulting "adaptive PCD" algorithm for future work, preliminary results (see Fig. 4c) suggest that one can indeed reach better accuracies this way, while the tradeoff (7) remains valid.

## Discussion

In summary, the accuracy–efficiency tradeoff is an inherent limitation of the RBM architecture and its reliance on Gibbs sampling (5) to assess the model distribution $\hat{p}_\theta(x)$. Depending on the eventual application of the trained model, this limitation should already be taken into account when planning and performing training: Aiming at higher accuracy implies that more resources will be required also in the production stage to evaluate and employ the trained model in an unbiased fashion.

Not least, the tradeoff directly affects the training process itself. It is well known that common training algorithms like contrastive divergence and its variants are biased[29,43] and that the bias increases with the magnitude of the weights[33,44]. Hence there exists an optimal stopping time for training at which the accuracy becomes maximal, but unfortunately, no simple criterion in terms of accessible quantities is known to determine this stopping time[44,45]. Approximate test errors like $\tilde{\Delta}_\sigma^{(T,T)}$, $\tilde{\delta}_\theta$ or $\tilde{\ell}_\theta^1$ can provide a rough estimate for when deterioration sets in, but are insensitive to finer details by construction. By contrast, taking the reconstruction error as a measure for the model accuracy, which is still not uncommon since it is easily accessible, is downright detrimental from a sampling-efficiency point of view because it decreases with increasing correlations between samples. Since it is not correlated with the actual loss either[44], the reconstruction error should rather be regarded as an efficiency measure (with larger "error" indicating higher efficiency).

The aforementioned fact that the magnitude of the weights is closely related to the autocorrelation time $\tau_\theta$ (see also Supplementary Note 5) provides a dynamical understanding of the bias in the sense that larger $\tau_\theta$ calls for more steps in the Markov chain (5) to obtain an effectively independent sample. Similar conclusions have been drawn from studies of the mixing time of RBM Gibbs samplers[27,33,35,36]. The mixing time quantifies how many steps in (5) are necessary to reach the stationary distribution $\hat{p}_\theta(x)$ from an arbitrary initial distribution for $x^{(0)}$. In CD training, where $x^{(0)}$ is taken from the training data (meaning that it is a sample drawn from $p(x)$ by assumption), it is particularly relevant for the early training stages when $\hat{p}_\theta(x)$ is possibly far away from the target. For analyzing a trained model, by contrast, the mixing time is less important because it only provides a constant offset to the sampling efficiency by quantifying the burn-in steps in (5), i.e., the number of samples to discard until the stationary regime is reached, whereafter one will start recording samples to actually assess $\hat{p}_\theta(x)$. Similarly, correlations in the PCD update steps are better described by autocorrelation times like $\tau_\theta$, at least if the learning rate is sufficiently small so that the Markov chains can be considered to operate in the stationary regime throughout training, and the same applies to ordinary CD updates at later training stages.

There are a variety of proposals to modify the sampling process so that correlations between subsequent samples in an appropriate analog of (5) are reduced, including the above-sketched PCD extension with $\tau_\theta$-adaptive order of the Markov-chain sampling (see also Fig. 4c), parallel tempering[26,32], mode-assisted training[46], or occasional Metropolis-Hastings updates[47,48]. However, these adaptations come with their own caveats and the extent to which correlations are reduced may depend strongly on the setting[35,48]. Moreover, the computational complexity of these methods is usually higher because additional substeps are necessary to produce a new Markov-chain sample. While a detailed quantitative analysis is missing, the overall evaluation efficiency (e.g., the required computational resources) will presumably not be improved in general[25], and probably the only remedy to circumvent the sampling problem could be novel computing hardware such as neuromorphic chips[49–53], "memcomputing machines"[54], or quantum annealers[55,56].

For a more comprehensive understanding of the tradeoff mechanism, it would be desirable to elucidate the role of the exponent $\alpha$ in (7) and how it relates to properties of the target distribution $p(x)$. As discussed above, $\alpha$ roughly quantifies how apt the RBM architecture is to represent $p(x)$, with larger values of $\alpha$ indicating better suitability. A related question is what distributions can be represented efficiently by RBMs in terms of the required number of hidden units[38,40,57]. Besides the number of "active" states, symmetries that make it possible to represent the correlations between various visible units with fewer hidden units could play an important role in affecting $\alpha$ (see also Supplementary Note 5). Furthermore, observing the marked transition from independent to correlation learning, one may naturally wonder whether there exists a hierarchy of how and when correlations are adopted during the correlation-learning regime[40,58–61], particularly

when $\alpha$ is ambiguous (e.g., in Fig. 2d; see also Supplementary Note 5). In any case, it is remarkable that in most of the examples we explored, $\alpha$ turns out to be approximately $\frac{1}{2}$, particularly at the initial stage of the correlation-learning regime. Whether this is a coincidence or a hint at some deeper universality principle is an intriguing open question.

# Methods

## Conditional RBM distributions

The approach to use alternating Gibbs sampling of visible and hidden units via Markov chains of the form (5) is viable in practice only due to the bipartite structure of the RBM with direct coupling exclusively between one visible and one hidden unit. Consequently, the visible units are conditionally independent given the hidden ones and vice versa, e.g., $\hat{p}_\theta(h|x) = \prod_j \hat{p}_\theta(h_j|x)$ with

$$\hat{p}_\theta(h_j|x) = \frac{e^{\left(\sum_i w_{ij}x_i + b_j\right)h_j}}{1 + e^{\sum_i w_{ij}x_i + b_j}}, \tag{9}$$

and similarly $\hat{p}_\theta(x|h)$ can be obtained by replacing $x_i \leftrightarrow h_j$ and $a_i \leftrightarrow b_j$ and by summing over $i$ in the exponents and taking the product over $j$. Sampling from $\hat{p}_\theta(h|x)$ and $\hat{p}_\theta(x|h)$ is thus of polynomial complexity in the number of units and can be carried out efficiently. Likewise, this explains why the first average on the right-hand side of (4) with $\tilde{p}(x; S)$ in lieu of $p(x)$ (sometimes called the "data average;" see also below Eq. (4)) can be readily evaluated. For $\theta_k = w_{ij}$, for example, one finds

$$\left\langle \frac{\partial E_\theta(x,h)}{\partial w_{ij}} \right\rangle_{\hat{p}_\theta(h|x)\tilde{p}(x;S)} = -\frac{1}{|S|} \sum_{x\in S} x_i\, \hat{p}_\theta(h_j=1|x), \tag{10}$$

and similarly for $a_i$ and $b_j$.

The variability of samples obtained from those conditional distributions can be assessed in terms of their Shannon entropy, defined for an arbitrary probability distribution $p(x)$ as $S(p) := -\sum_x p(x)\log p(x)$. Specifically,

$$S(\hat{p}_\theta(h|x)) = \sum_j \left[ \log\left(1 + e^{\sum_i w_{ij}x_i + b_j}\right) - \frac{\sum_i w_{ij}x_i + b_j}{1 + e^{-\sum_i w_{ij}x_i - b_j}} \right], \tag{11}$$

and, again, similarly for $\hat{p}_\theta(x|h)$. The entropy is maximal for the uniform distribution with $\theta_k = 0$ for all parameters. It remains large as long as the $\theta_k$'s are small in magnitude and tends to decrease towards zero as $|\theta_k|$ increases unless there is a special fine-tuning for specific configurations $h$ that leads to exact cancelations. Over multiple steps of the Markov chain (5), the samples will thus generically show more variability for small weights, whereas they develop stronger correlations as the weights grow[33,44] (see also Supplementary Note 5).

## Details on $\Delta_\theta$, $C_{\text{tot}}(\hat{p}_\theta)$ and related quantities

The measure of accuracy $\Delta_\theta$ (exact loss, ideal test error) is calculated numerically exactly by carrying out the sums in Eqs. (2) and (3). Similarly, the total correlations $C_{\text{tot}}(p)$ of the target and model distributions are computed exactly according to (8) as a sum over all states that keeps track of the contributions from both the full distribution $p(x)$ and the marginal ones $p_i(x_i)$.

For the partition function (2), we can exploit the bipartite structure of the RBM's interaction graph, such that one of the sums can be factorized and thus be evaluated efficiently. For example, if $N \leq M$, we rewrite (2) as

$$Z_\theta = \sum_h e^{\sum_j b_j h_j} \prod_i \left(1 + e^{\sum_j w_{ij}h_j + a_i}\right), \tag{12}$$

and similarly if $M < N$. The sum over $h$ in (12) involves $2^N$ terms, but the product over $i$ in each summand consists of just $M$ factors. Therefore,

the computational complexity scales exponentially with $\min\{M,N\}$ only. For the sum in Eq. (3), we can exploit the sparsity of the target distribution $p(x)$ and restrict the (costly) evaluations of $\hat{p}_\theta(x)$ to those states with $p(x) > 0$. Notwithstanding, the system sizes for which the computation of $\Delta_\theta$ remains viable is relatively small; see also refs. 26,44–46,62 for studies of the exact RBM loss in small examples.

In practical applications, one does not have access to $p(x)$, but only to a collection of samples $S := \{\tilde{x}^{(1)}, \ldots, \tilde{x}^{(|S|)}\}$ (training and/or test data). The empirical counterpart of $\Delta_\theta$ for such a dataset $S$ is

$$\tilde{\Delta}_\theta^{(S)} = -\frac{1}{|S|} \sum_{x\in S} \left[ \sum_i a_i x_i + \sum_j \log\left(1 + e^{\sum_i w_{ij}x_i + b_j}\right) \right] + \log Z_\theta - \log|S|; \tag{13}$$

see also below Eq. (4). The critical part is again the partition function $Z_\theta$. Due to the aforementioned factorization (cf. Eq. (12)), evaluating (13) remains feasible as long as the number of hidden units $N$ is sufficiently small, even if $M$ is large. Similarly, for small $N$, we can draw independent samples from $\hat{p}_\theta(x) = \sum_h \hat{p}_\theta(x|h)\hat{p}_\theta(h)$, without reverting to Markov chains and Gibbs sampling: We first generate independent samples $\{\tilde{h}^{(\mu)}\}$ of the hidden units, using the fact that $\hat{p}_\theta(h)$ remains accessible for small $N$. Subsequently, we sample configurations of the visible units using $\hat{p}_\theta(x|h = \tilde{h}^{(\mu)})$. This scheme was utilized to obtain the model test samples $\hat{T}$ for the $N \leq 32$ examples in Fig. 4. For the examples with $N > 32$, the samples in $\hat{T}$ were instead generated via Gibbs sampling according to (5), using 10 parallel chains and storing every $\tau_\theta$-th sample after $2 \times 10^6$ burn-in steps.

The accuracy measures $\tilde{\delta}_\theta$ and $\tilde{\ell}_\theta^{-1}$ involve empirical distributions of coarse-grained visible-unit samples. These reduced samples are obtained by using a weighted majority rule: For a partition $\{L_1, \ldots, L_L\}$ of the visible-unit indices $\{1, \ldots, M\}$ and a threshold $r \in [0, 1]$, we define

$$f_\alpha(x) := \begin{cases} 1 & \text{if } \sum_{i\in L_\alpha} x_i \geq r\,|L_\alpha|; \\ 0 & \text{otherwise}. \end{cases} \tag{14}$$

For every sample $\tilde{x}$ in a given multiset $S$, the associated coarse-grained sample is $\tilde{y} = (\tilde{y}_1, \ldots, \tilde{y}_L)$ with $\tilde{y}_\alpha := f_\alpha(\tilde{x})$.

## Details on $\tau_\theta$

To measure the efficiency of Gibbs sampling according to the Markov chain (5), we evaluate the integrated autocorrelation time $\tau_\theta$ from (6). The general purpose of Gibbs sampling is to estimate the model average $\langle f(x)\rangle \equiv \langle f(x)\rangle_{\hat{p}_\theta(x)}$ of some observable $f(x)$, i.e., a function of the visible units. The sample mean $\bar{f} := \frac{1}{R}\sum_{n=0}^{R-1} f(x^{(n)})$ over a chain of $R$ samples is an unbiased estimator of $\langle f(x)\rangle$ if the chain is initialized and thus remains in the stationary regime, $x^{(0)} \sim \hat{p}_\theta(x)$ (see also below Eq. (6)). The correlation function associated with $f(x)$ and the Markov chain (5) is

$$g_\theta^{(f)}(n) := \langle f(x^{(0)})f(x^{(n)})\rangle - \langle f(x)\rangle^2. \tag{15}$$

For any such correlation function $g_\theta^{(f)}(n)$, the corresponding integrated autocorrelation time is defined similarly to Eq. (6),

$$\tau_\theta^{(f)} := 1 + 2\sum_{n=1}^\infty \frac{g_\theta^{(f)}(n)}{g_\theta^{(f)}(0)}. \tag{16}$$

To assess the reliability of the estimator $\bar{f}$, we inspect its variance

$$\langle \bar{f}^2\rangle - \langle\bar{f}\rangle^2 = \frac{g_\theta^{(f)}(0)}{R}\left[1 + 2\sum_{n=1}^{R-1}\left(1 - \frac{n}{R}\right)\frac{g_\theta^{(f)}(n)}{g_\theta^{(f)}(0)}\right]. \tag{17}$$

If the number of samples $R$ is much larger than the decay scale of $g_\theta^{(f)}(n)$ with $n$ (which is a prerequisite for estimating $\bar{f}$ reliably), the

contribution proportional to $\frac{n}{R}$ becomes negligible in the sum and the term in brackets reduces to $\tau_\theta^{(f)}$ from (16); see also Sec. 2 of ref. 34. Observing that $g_\theta^{(f)}(0)$ is the variance of $f(x)$, the variance of the estimator $\bar{f}$ from correlated Markov-chain samples is thus a factor of $\tau_\theta^{(f)}$ larger than the variance of the mean over independent samples. In other words, sampling via the Markov chain (5) requires $\tau_\theta^{(f)}$ more samples than independent sampling to reach the same standard error and is thus less efficient the larger $\tau_\theta$ becomes.

In general, the integrated autocorrelation times $\tau_\theta^{(f)}$ can and will be different for different observables $f(x)$. The specific choice $\tau_\theta$ from (6) is supposed to capture the generic behavior of typical observables. It focuses on the individual visible units $x_i$ as the elementary building blocks. However, instead of taking the mean over the autocorrelation times $\tau_\theta^{(x_i)}$ for each unit $f(x) = x_i$, the averaging is performed at the level of the correlation functions $g_\theta^{(x_i)}(n)$; cf. below Eq. (6). The effect is a weighted average

$$\tau_\theta = \frac{\sum_i g_\theta^{(x_i)}(0)\,\tau_\theta^{(x_i)}}{\sum_i g_\theta^{(x_i)}(0)} \tag{18}$$

that gives higher importance to strongly fluctuating units with a large variance $g_\theta^{(x_i)}(0)$. This accounts for the fact that variability of the Markov-chain samples is more important for those units and reduces the risk of underestimating correlations when there are certain regions in the data that behave essentially deterministically, e.g., background pixels at the boundary of an image distribution.

In practice, if one is interested in a specific observable $f(x)$, the associated autocorrelation time $\tau_\theta^{(f)}$ should be monitored directly instead of (or along with) the generic $\tau_\theta$. While the quantitative details may differ, we expect that the scaling behavior and the tradeoff mechanism remain qualitatively the same. A comparison for different observables in the TFIC example from Fig. 2 and in the digit-pattern images from Fig. 4a–c can be found in Supplementary Note 4. We indeed observe that $\tau_\theta^{(f)}$ is usually largely proportional to $\tau_\theta$.

In our numerical experiments, we estimate $\tau_\theta$ statistically from long Markov chains of the form (5) with $n_{tot}$ samples. Due to sampling noise, the sum over time lags $n$ in (6) must be truncated at a properly chosen threshold $n_{max}$ to balance the bias and variance of the estimator. Following ref. 34, we choose $n_{max}$ as the smallest integer such that $n_{max} \geq \gamma\,\tilde{\tau}_\theta(n_{max})$, where $\gamma$ is a constant and $\tilde{\tau}_\theta(n_{max})$ is the value obtained from truncating (6) at $n_{max}$ using empirical averages to estimate the correlation function $g_\theta(n)$ (see below Eq. (6)) and exploiting translational invariance of the stationary state (i.e., $\langle x_i^{(0)} x_i^{(n)} \rangle = \langle x_i^{(k)} x_i^{(n+k)} \rangle$). If $g_\theta(n)$ follows an exponential decay, the bias of the estimator is of order $e^{-\gamma}$, and we use $\gamma = 5$ in Figs. 2 and 3 and $\gamma = 8$ in Fig. 4. To reach the stationary regime, we initialize the chain (5) in a state sampled uniformly at random and thermalize it by discarding a large number of samples, at least on the order of $100\tau_\theta$, providing a reasonable buffer to account for mixing times that may exceed $\tau_\theta$ (and would thus increase the bias if the number of discarded samples was too small).

In Fig. 4, we additionally maintain $r_g$ independently initialized chains to estimate $g_\theta(n)$ and calculate $\tau_\theta$ as described above, using the average over the $r_g$ chains for $g_\theta(n)$. The estimates are considered to be reliable only if the variations between the means of the $r_g$ chains are below 5 %; otherwise the data points are discarded. Furthermore, we repeat the entire procedure $r_\tau$ times, leading to $r_\tau$ independent estimates of $\tau_\theta$. The error bars in Fig. 4 indicate the min-max spread between those $r_\tau$ estimates.

## Power-law bound

In the examples from Figs. 2–4, the blue dashed lines indicate the power-law bound (7) for the accuracy–efficiency tradeoff. The constants $c$ and $\alpha$ in this bound as stated in the respective figure panels were determined as follows: The exponent $\alpha$ is chosen to roughly match the average slope $-\frac{\partial \log \Delta_\theta}{\partial \log \tau_\theta}$ for the data points in the correlation-learning regime over all hyperparameter configurations ($n_{CD}$, $\eta$, $B$, $|S|$) for any specific target distribution $p(x)$. If this choice is ambiguous (e.g., in Fig. 2d), the behavior in the beginning of the correlation-learning regime ($\tau_\theta \simeq 1$, $\Delta_\theta \simeq C_{tot}(p)$) is decisive. Once $\alpha$ is fixed, $c$ is chosen as the maximum value such that $\Delta_\theta \tau_\theta^\alpha \geq c$ holds for all data points of all hyperparameter configurations simultaneously.

## Examples

The first examplary task (cf. Fig. 2) is quantum-state tomography, namely to learn the ground-state wave function of the TFIC based on measurements of the magnetization in a fixed spin basis $\{|x_1 \cdots x_M\rangle\}$, where $x_i = 0$ ($x_i = 1$) indicates that the $i$th spin points in the "up" ("down") direction in the chosen basis. The Hamiltonian is $H = -\frac{1}{2}\sum_{i=1}^M (\sigma_i^x \sigma_{i+1}^x + g\,\sigma_i^z)$ with periodic boundary conditions and Pauli matrices $\sigma_i^\gamma$ ($\gamma = x, y, z$) acting on site $i$. The model exhibits a quantum critical point at $|g| = 1$ and is integrable, such that the ground state $|\psi\rangle = \sum_x \psi(x)|x_1 \cdots x_M\rangle$ can be constructed explicitly[63,64] (see also Supplementary Note 2A). As we consider measurements in the $\sigma^z$ and $\sigma^x$ directions only, the basis states $|x_1 \cdots x_M\rangle$ can be chosen such that $\psi(x)$ is real-valued and non-negative, which allows us to employ the standard RBM architecture (1). (Generalizations for complex-valued wave function are possible[10,51]). The target distribution is thus $p(x) = \psi(x)^2$.

Our second example (cf. Fig. 3) is closer in spirit to traditional machine-learning applications and involves pattern recognition and artificial image generation. The target distribution $p(x)$ generates $5 \times 5$ pixel images with a "hook" pattern comprised of 15 pixels (see Fig. 3a) implanted at a random position in a background of noisy pixels that are independently activated (white, $x_i = 1$) with probability $q = 0.1$ (see also Supplementary Note 2B for more details). Periodic boundary conditions are assumed, meaning that $p(x)$ is translationally invariant along the two image dimensions.

We also consider a one-dimensional variant of this example with only $M = 4$ ($M = 5$) visible units and an implanted "010" ("0110") pattern, cf. Fig. 3d. In this case, we can solve the continuous-time learning dynamics ($\eta \to 0$ limit of (4)) for the exact target and model distributions $p(x)$ and $\hat{p}_\theta(x, h)$, obviating artifacts caused by insufficient training data or biased gradient approximations, see also Supplementary Note 1.

Our third example (cf. Fig. 4a–c) is a simplified digit reproduction task. Patterns of the ten digits 0 through 9 (see Fig. 4a) are selected and inserted uniformly at random into image frames of $5 \times 7$ pixels, with the remaining pixels outside of the pattern again activated with probability $q = 0.1$ (see Supplementary Note 2C for details). No periodic boundary conditions are imposed, i.e., the input comprises proper, ordinary images.

In our fourth example (cf. Fig. 4d, e), we train RBMs on the MNIST dataset[42], which consists of $28 \times 28$-pixel grayscale images of handwritten digits. It comprises a training set of 60,000 and a test set of 10,000 images. We convert the grayscale images with pixel values between 0 and 255 to binary data by mapping values 0…127 to 0 and 128…255 to 1 (see also Supplementary Note 2D).

## Data availability

The source data of Figs. 2–4 are provided with this paper in the Source Data file. Owing to the large file size of the full dataset, the raw data that support the findings of this study are available as needed from the corresponding author upon reasonable request. Source data are provided with this paper.

## Code availability

The computer code for the numerical experiments can be accessed from the public repository https://gitlab.com/lennartdw/xminirbm.

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

## Acknowledgements

This work was supported by KAKENHI Grant No. JP22H01152 from the Japan Society for Promotion of Science.

## Author contributions

L.D. carried out the calculations and simulations. L.D. and M.U. discussed and interpreted the results and wrote the manuscript.

## Competing interests

The authors declare no competing interests.
