## [Peer Review File · Nature Communications]

REVIEWER COMMENTS

Reviewer #1 (Remarks to the Author):

The authors of this manuscript study the different training stages of a restricted Boltzmann machine (RBM). In order to do so they distinguish between the accuracy and the efficiency of the RBM, where the efficiency is considered based on correlations between samples drawn from the RBM via a Markov chain. The longer the autocorrelation times are, the more sampling is necessary to get a desired number of independent samples, which makes the algorithm computationally less efficient. Three different learning stages are observed, where first the accuracy improves at a constant efficiency, where it is shown that the optimal accuracy with independent sampling is reached, before both efficiency and accuracy improve. At the latest third stage, the accuracy typically gets worse, so that training should be stopped before this stage.

The authors demonstrate that this behaviour can be observed in three different examples for RBM applications, where they consider quantum state reconstruction of the transverse-field Ising model, a pattern generation task, and digit generation. With this they cover a great variety of famous RBM applications, demonstrating the generalizability of their results while focussing on small RBMs with clearer interpretability.

The manuscript is very well structured and nicely written. All ingredients and details are explained and introduced well, which makes it easy to follow the individual steps. All necessary background information is provided and conclusions are drawn based on a solid foundation. The topic is very recent and provides a great impact in the field of RBMs, as their exact behaviour is far from well understood and requires more insights. Since RBMs nowadays find applications in a wide field of topics, this step towards a better understanding of their performance is important for many researchers. I thus recommend publication of this paper in Nature Communications. However, the authors should consider the following improvements of the manuscript before acceptance:

- The figure setup was sometimes slightly confusing. The authors should ensure that text and data points do not overlap as the text is hard to read otherwise.

- Figure 1: Why are there ticks on the y-axis? Does the accuracy always increase by an order of magnitude? Since this is a sketch, I found this confusing.

- Figures 2 and 3: I found it hard to distinguish the different data points and lines, as many of them overlap very often. Is it possible to make this more clear, eg by decreasing the size of some data points?

- Figures 2, 3, and 4: All data points are averaged over five RBM runs. What are the variances and error bars? Information on this is provided in the supplement, but it should also be available in the main text.

- Figure 4 contains a lot of information. I was first confused why there are multiple lines with the same colors/parameters in a single patch, until I figured out that other parameters are varied for those. The authors should explain this plot more clearly in the figure caption or maybe think about a way to make the plots clearer (even though I don't have an idea how to do this either).

- Line 91: The authors state that the first average in Eq. (4) can readily be evaluated. How is $p_{\theta}(x|h)$ evaluated? I found this information in more detail in the Methods section, but it was hard to follow the argumentation here.

- Line 112: What happens if δ_{θ} is not sufficiently small, as it is the case at the beginning of the training process? How is the Markov chain created if the first sample cannot be considered as an approximation of a sample from $p_{\theta}(x)$?

- Line 149: The authors state that the autocorrelation times of different operators are expected to scale similarly. Why is this the case? Autocorrelation times can vary heavily for different operators, why should their scaling not vary? In this context it would be interesting to see if the training stages show a similar behaviour for different operators.

- Eq. (8): How is the product distribution of independent units determined in the denominator of $C_{\text{tot}}(p)$? I don't see how it is straight-forward to get expressions for the distributions of the individual units from the full distribution. Are those just extracted from samples?

- Line 211: What does the intersection of the red and blue dashed lines in the figures illustrate? How is α chosen, which influences this intersection?

- Line 227: The weights in an RBM are never exactly zero, so that C_{tot} is only reached within limits. What is the scale here, when do the authors say that C_{tot} is reached and how large are the weights for these cases? For what weights are the neurons considered as uncorrelated?

- Line 261: What is the Shannon entropy and what does it tell us here? Again I found the information later in the Methods section, but it makes it hard to follow the paper at this point.

- Line 317: I don't see how the accuracy improvement is more clear for larger B. It looks similar for me in all panels of Fig. 3b, the authors should explain where they see this improvement.

- Generally: How are the RBM weights initialized? This is briefly discussed in the supplement, but I would think that this highly influences the training stages of the RBM. What happens if we start with larger initial weights, which would result in initial correlations between the neurons? How small do the weights need to be chosen so that we consider the neurons as initially uncorrelated? Since the weights are not initialized as exactly zero, there are some weak initial correlations present.

- In the scope of references [45-47] the authors could as well cite S. Czischek et al., doi:10.21468/SciPostPhys.12.1.039 and R. Klassert et al., arXiv:2109.15169, which show first demonstrations of using spiking neuromorphic hardware to sample from quantum state representations.

Reviewer #2 (Remarks to the Author):

Referee Report for "Three Learning Stages and Accuracy-Efficiency Tradeoff of Restricted Boltzmann Machines"

This work studies the curve in the Δ - τ plane that is traced during training a binary restricted Boltzmann machine. Here Δ is the KL-divergence from a target distribution to the distribution represented by the RBM and τ is the integrated autocorrelation time. It is proposed and observed in a series of experiments that there are three stages: one where the margins of individual variables are adjusted, one where interactions between variables are adjusted, and one of degradation. It is observed that there is an accuracy-efficiency tradeoff, whereby $\Delta \tau^\alpha$ is bounded below.

I found this to be a refreshing view on RBMs. The discussion of learning stages is an interesting and contemporary topic. The presentation of the learning stages in a Δ - τ plane appears to be new and I found it interesting, since τ could be related to sampling efficiency. The text is mostly well written.

My main disappointment with the submission is that

1) the observations about the three learning stages are more or less expected, at least at a high level. I think the contribution could be more impactful if it included clearer recommendations or conclusions derived from the analysis.

2) there are no theoretical results being presented to substantiate the high-level intuitions,

3) the experiments that are being presented in order to substantiate the claims are relatively simple.

In regard to 1): of course a simple intuition that is substantiated with theory and that leads to useful practical recommendations can make a strong paper. The paper would be stronger if made more advances in this direction.

In regard to 2): there is a growing number of works investigating the training dynamics of neural networks, both theoretically and experimentally. Hence I anticipated that some theoretical advances might form part of the present contribution for RBMs. Of course not all papers must include new theory or strong technical results, but in the case that strong theory is not part of the paper I believe the the experimental part would need to be accordingly stronger.

In regard to 3): the experimental section focuses on relatively simple settings. This is to some extent justified since evaluating large RBMs is difficult. However, I think the contribution would have been stronger if it included experiments addressing (some of) the following questions:

- what happens for larger RBMs (do the observations for small RBMs extrapolate to large models / complex target distributions), can this be illustrated in terms of estimated / approximated versions of Δ and τ ?
- what is the impact of the parameter initialization procedure on the regimes being presented?
- what is the impact of the learning rate and other details of the learning algorithm (the paper does include comparison of the mini batch size and order of CD)?
- can the behaviour of τ be compared with the sampling complexity and their relation verified / evaluated experimentally?
- can one obtain a convergence rate result for the training error / generalization error, or describe the training error / generalization error against the training data / computation time / computation effort?
- can one obtain a more explicit description of the qualitative differences in the Δ - τ curves depending on the type of target distribution / size of the RBM?

Other comments

* line 138 ``we would then need on the order of $\tau\theta R$ correlated Markov-chain samples [33]" Explain why and indicate specific statement in the referenced work.

* Section Mechanism. Here a discussion of the initialisation is needed.

* In line 202, add explanation for why the decomposition $D(p||\hat{p})=C(p)+D(p||\hat{p})$ is valid. This being claimed to then conclude the weaker statement $\Delta \geq C_{\text{tot}}(p)$.

* line 214 ``no significant correlations between the RBM's visible units build up as long as $\tau_{\theta} \approx 1$ "

Is this not true by definition?

* line 222 ``whose standard deviation" This is defined later but would be good to define it by the time it appears first in the next.

* line 243 `` σ^x " add definition here

* Fig 2. definition of σ^z and σ^x basis missing. ``(b) ... points" points are not visible in the plot (this could be more visible if reducing the size). ``solid lines" these are also not visible. ``All data points correspond to averages over 5 independent training runs" This could be explained more clearly.

* Fig 3. is also difficult to read.

* line 336 ``when the accuracy of the model approaches that of the empirical distribution" This probably needs rephrasing, since it makes no sense as stated.

* Fig 4. "50000 samples" Here it would be good to explain more clearly what is the variability of the samples, since most pixels are occupied by a small number of different patterns.

* line 454 "what distributions can be represented efficiently by RBMs... symmetries that make it possible to represent the correlations" Here two references appear useful to mention, which explain how correlations among visible units are represented in RBMs and the types of interactions that can be represented by hidden units connected with visible units:

- Younes. Synchronous Boltzmann machines can be universal approximators. Applied Mathematics Letters, 9(3):109–113, 1996.

- Montufar and Rauh. Hierarchical Models as Marginals of Hierarchical Models. International Journal of Approximate Reasoning ; DOI: 10.1016/j.ijar.2016.09.003; Date: September, 2016.

* line 487 "decreases towards zero as $|\theta_k|$ increases". Although this tends to happen, the statement is not exactly correct. It is possible to have all individual parameters have parameters with magnitude going to infinity without the entropy going to zero.

* line 578 "Data availability... upon reasonable request" I would suggest that a supplementary material with the computer code to replicate the experiments be submitted with the paper.

* Another article that appears relevant in the context of this work is:

Shun-ichi Amari, Information Geometry on Hierarchy of Probability Distributions, IEEE Transactions on Information Theory 47(5):1701 - 1711, August 2001.

Reply to Reviewer Reports

We are very thankful to both reviewers for their thorough evaluation of our work and their highly insightful comments.

The major changes and amendments in response to those comments are:

- we develop a scheme to assess the tradeoff between accuracy and efficiency approximately in larger systems (new section “Towards applications”) and exemplify it by means of the prototypical MNIST dataset (new Fig. 4);
- we carry out a finite-size analysis for the case of the transverse-field Ising model (Fig. 2), which has a natural scaling limit for the number of visible units (= lattice sites), to reinforce that the three learning stages and the accuracy–efficiency tradeoff are not an artifact of small systems and are to be expected similarly in practical applications;
- we formulate concrete recommendations for practical applications (new section “Towards applications”) and suggest an extension to the persistent contrastive divergence (PCD) algorithm that should allow one to obtain higher-accuracy models (albeit with larger evaluation cost, as predicted by our tradeoff);
- we extend the discussion of the quantity τ_θ , of integrated autocorrelation times in general, and of their relation to the sampling efficiency (sections “Accuracy and efficiency” and “Methods”), including examples to illustrate variations for different observables (see also Supplementary Note S4).

To make space for the amendments, parts of the arguments and discussion in “Mechanism” have been shortened and/or moved to the Supplementary Information (Sec. S5 in particular). We have color-coded all text changes.

In the following, we reply point by point to the reviewer comments.

Reviewer 1

We thank the reviewer for his/her positive evaluation of our work and the valuable suggestions for improvement.

- The figure setup was sometimes slightly confusing. The authors should ensure that text and data points do not overlap as the text is hard to read otherwise.

We reproduced all figures (see also below) and made sure that there is no overlap between text and data points.

- Figure 1: Why are there ticks on the y-axis? Does the accuracy always increase by an order of magnitude? Since this is a sketch, I found this confusing.

The ticks were originally included to indicate the log-scaling of the axes. We understand that this can be confusing and we have removed the ticks (on both axes). Instead, we now indicate the log-scaling in the axes labels.

- Figures 2 and 3: I found it hard to distinguish the different data points and lines, as many of them overlap very often. Is it possible to make this more clear, eg by decreasing the size of some data points?

A similar request was brought forward by Reviewer 2 as well. We have decreased the size of the data points. To make the points even smaller, we would need to sacrifice information that is encoded in the filling and border colors in Figs. 2 and 3 (filling color relates to $C_{\text{tot}}(\hat{p}_\theta)$, border color relates to number of hidden units N). For now, we have therefore decided to add an additional comment in the main text that explains the visualization in more detail (see lines 193–198). However, we are willing to reconsider discarding information if the reviewers still perceive that the present visualization is not clear enough.

Please also note that we sometimes deliberately display data points that overlap because this highlights the independence of the learning curves from the hyperparameters. More precisely, in some figures we show learning characteristics (Δ_θ vs. τ_θ) for different values of a hyperparameter in the same panel. If those curves overlap, it shows that the relationship between Δ_θ and τ_θ is insensitive to variations of that hyperparameter.

- Figures 2, 3, and 4: All data points are averaged over five RBM runs. What are the variances and error bars? Information on this is provided in the supplement, but it should also be available in the main text.

We have moved and expanded the information about repetitions and uncertainties from the supplement to the main text (see beginning of “Mechanism” as well as “Methods”).

- Figure 4 contains a lot of information. I was first confused why there are multiple lines with the same colors/parameters in a single patch, until I figured out that other parameters are varied for those. The authors should explain this plot more clearly in the figure caption or maybe think about a way to make the plots clearer (even though I don’t have an idea how to do this either).

This figure has been moved to the Supplementary Information (Fig. S7) with a more detailed caption to highlight the figure’s main message.

- Line 91: The authors state that the first average in Eq. (4) can readily be evaluated. How is $p_\theta(x|h)$ evaluated? I found this information in more detail in the Methods section, but it was hard to follow the argumentation here.

We have added an explicit expression, Eq. (10), showcasing the form that averages like the first term on the right-hand side of Eq. (4) take. Together with Eq. (9), this explains how those averages can be evaluated efficiently (i.e., in polynomial time in M, N).

- Line 112: What happens if Δ_θ is not sufficiently small, as it is the case at the beginning of the training process? How is the Markov chain created if the first sample cannot be considered as an approximation of a sample from $p_\theta(x)$?

By definition, the contrastive divergence (CD) algorithm always chooses a sample of the training data to initialize the chain. As pointed out rightfully by the reviewer, this is less justified in the beginning of training when the model distribution differs strongly from the target. Nevertheless, this approach seems to facilitate learning in practice. Understanding the bias of CD and its effect on the learning progress is a vast research topic on its own; see, for instance, Refs. [23, 26, 28, 32, 42, 43]. A detailed discussion of this subject is beyond the scope of our present manuscript. We merely use CD in most of our examples because it is the most common training algorithm for RBMs.

There are alternative schemes for the Markov-chain sampling during training, notably persistent CD (PCD) and its variants, cf. Refs. [29, 30]. Here, the final state from the previous training step is used as the initial state for the chain in each step. In the revised manuscript, we also show results using

this approach. It can sometimes allow us to reach better accuracies (smaller Δ_θ), but crucially it cannot beat the accuracy–efficiency tradeoff (τ_θ increases accordingly if Δ_θ becomes smaller). This reinforces the observation that our main quantities of interest, Δ_θ and τ_θ , are intrinsic properties of the RBM distribution and the principal origin of the tradeoff is the encoding of correlations between visible units via the hidden ones, which will likewise happen in any other (successful) approximate training scheme. Furthermore, in the example from Fig. 3e ($n_{\text{CD}} = \infty$), training is carried out exactly according to Eq. (4) (without approximations), and we still observe a similar behavior.

We decided to remove the remark addressed by the reviewer from the main text to avoid confusion. Instead, we briefly comment on this issue in Supplementary Note S1 now (see below Eqs. (S2) and (S3)).

- Line 149: The authors state that the autocorrelation times of different operators are expected to scale similarly. Why is this the case? Autocorrelation times can vary heavily for different operators, why should their scaling not vary? In this context it would be interesting to see if the training stages show a similar behaviour for different operators.

This is an important comment that led us to elaborate in more detail on the autocorrelation times in the “Accuracy and efficiency” and “Methods” sections and to include additional simulation results for different operators in the Supplementary Information (Sec. S4).

- Eq. (8): How is the product distribution of independent units determined in the denominator of $C_{\text{tot}}(p)$? I don’t see how it is straight-forward to get expressions for the distributions of the individual units from the full distribution. Are those just extracted from samples?

Similarly to Δ_θ , the total correlation $C_{\text{tot}}(p)$ or $C_{\text{tot}}(\hat{p}_\theta)$ was evaluated numerically exactly. The calculation scheme is “brute force,” we cycle all states in Eq. (8) explicitly and keep track of their contributions to the individual distributions $p(x)$ and $p_i(x_i)$. Of course, this approach is viable only for small numbers of visible units. We slightly extended the relevant paragraph in “Methods” where the calculation of $C_{\text{tot}}(p)$ is addressed.

- Line 211: What does the intersection of the red and blue dashed lines in the figures illustrate? How is α chosen, which influences this intersection?

We apologize that this important information was missing. It is now provided in the “Methods” section (paragraph “Power-law tradeoff”). The parameters c and α in Eq. (7) that determine the blue dashed lines in the figures are chosen such that

- α reflects the behavior of the majority of data points in the correlation-learning regime, or, if this is ambiguous such as in Fig. 2d, in the beginning of the correlation-learning regime;
- c is as large as possible (given α) such that Eq. (7) lower-bounds all data points for all displayed hyperparameter values simultaneously.

The red-dashed line indicates the value of $C_{\text{tot}}(p)$. Hence the intersection of the red and blue dashed lines (at $\tau_\theta = 1$, cf. the black dashed line) illustrates that c , the minimal loss achieved with independent units (when $\tau_\theta \simeq 1$), coincides with $C_{\text{tot}}(p)$, the total correlation of the target distribution, i.e., $c \simeq C_{\text{tot}}(p)$.

- Line 227: The weights in an RBM are never exactly zero, so that C_{tot} is only reached within limits. What is the scale here, when do the authors say that C_{tot} is reached and how large are the weights for these cases? For what weights are the neurons considered as uncorrelated?

In general, this is a subtle issue. For the visible units to pass as (nearly) independent or uncorrelated, their total correlation $C_{\text{tot}}(p)$ should certainly be much smaller than the total correlation $C_{\text{tot}}(p)$ of the target distribution. This sets an absolute scale. In particular, the initial values of the model

parameters should be so small that this is satisfied and that the weak remnant correlations can be corrected for within a few training steps. We comment on this issue in our new paragraph about parameter initialization (beginning of “Mechanism”).

This way, the visible units can be considered as “uncorrelated” in the beginning of training. Regarding the question of *how long* they can still be regarded as uncorrelated during training, it turns out that the transition from the independent-learning regime to the correlation-learning regime is quite marked in practice. On the one hand, this is reflected in the kink of the Δ_θ vs. τ_θ curves. On the other hand, similar kinks are observed when plotting τ_θ , $C_{\text{tot}}(\hat{p}_\theta)$, or even σ_w (magnitude of the weights) against the number of training epochs t , see for example Fig. 3c. From a practical point of view, it therefore seems to be quite evident when the units cease to be uncorrelated, and it is accompanied by a marked change in the scaling of the magnitude of the weights with the training time.

- Line 261: What is the Shannon entropy and what does it tell us here? Again I found the information later in the Methods section, but it makes it hard to follow the paper at this point.

In the revised manuscript, this issue is exclusively discussed in the “Methods” section (see around Eq. (11)), where the Shannon entropy is properly introduced.

- Line 317: I don’t see how the accuracy improvement is more clear for larger B . It looks similar for me in all panels of Fig. 3b, the authors should explain where they see this improvement.

This passage was formulated misleadingly. In fact, we merely wanted to draw the reader’s attention to the “stable” learning regime, i.e., to the subpanels with $B \geq 50$ (and perhaps also $B = 20, \eta = 0.5/B$). We adjusted the wording accordingly. We agree that there is no essential difference of the accuracy improvement with N between different B values in this regime (see also the comment around lines 268–300 regarding dependence on hyperparameters).

- Generally: How are the RBM weights initialized? This is briefly discussed in the supplement, but I would think that this highly influences the training stages of the RBM. What happens if we start with larger initial weights, which would result in initial correlations between the neurons? How small do the weights need to be chosen so that we consider the neurons as initially uncorrelated? Since the weights are not initialized as exactly zero, there are some weak initial correlations present.

We have moved the information about the initial conditions from the supplement to the main text (see beginning of “Mechanism”) and expanded the discussion to address the above questions. We also provide examples for other initializations, including ones with significant correlations, in the new Supplementary Note S3.

- In the scope of references [45–47] the authors could as well cite S. Czischek et al., doi :10.21468/SciPostPhys.12.1.039 and R. Klassert et al., arXiv:2109.15169, which show first demonstrations of using spiking neuromorphic hardware to sample from quantum state representations.

We thank the reviewer for pointing us to those references. We have added them in the suggested place.

Reviewer 2

We thank the reviewer for clearly stating his/her opinion and reservations and for very detailed proposals how to extend the analysis.

My main disappointment with the submission is that

1) the observations about the three learning stages are more or less expected, at least at a high level. I think the contribution could be more impactful if it included clearer recommendations or conclusions derived from the analysis.

In regard to 1): of course a simple intuition that is substantiated with theory and that leads to useful practical recommendations can make a strong paper. The paper would be stronger if made more advances in this direction.

This is a very helpful observation because we think that, in fact, our results suggest a variety of ways to estimate the model quality and accessibility also in practical applications, but we agree that we did not work those out very well in the previous manuscript. Moreover, the additional analysis carried out for this revision allows us to make more specific recommendations, too. We have added the new section “Towards applications” and amended the “Discussion” to include concrete suggestions to assess the accuracy–efficiency tradeoff in applications.

2) there are no theoretical results being presented to substantiate the high-level intuitions,

In regard to 2): there is a growing number of works investigating the training dynamics of neural networks, both theoretically and experimentally. Hence I anticipated that some theoretical advances might form part of the present contribution for RBMs. Of course not all papers must include new theory or strong technical results, but in the case that strong theory is not part of the paper I believe the the experimental part would need to be accordingly stronger.

We absolutely agree that it would be highly desirable to obtain mathematically rigorous insights into the learning dynamics of RBMs. On the other hand, the very reason why our present results are relevant is that basic properties of the RBM model (e.g., the loss function, the learned distribution) have so far defied direct evaluation in practice. In other words, we believe that our analysis of the sampling efficiency and its relation to the model accuracy is important *because* it is notoriously hard to characterize RBMs analytically, so it is all the more vital to explore how efficiently they can be characterized and employed numerically. To work out the practical relevance more clearly, we have substantially extended the experiments, as detailed below.

3) the experiments that are being presented in order to substantiate the claims are relatively simple.

In regard to 3): the experimental section focuses on relatively simple settings. This is to some extent justified since evaluating large RBMs is difficult. However, I think the contribution would have been stronger if it included experiments addressing (some of) the following questions:

We are thankful for the reviewer’s concrete suggestions and picked up most of them in the revised manuscript:

- what happens for larger RBMs (do the observations for small RBMs extrapolate to large models / complex target distributions), can this be illustrated in terms of estimated / approximated versions of Δ and τ ?

There are two amendments in the revision that explicitly address the behavior for larger systems:

- In the example of ground-state tomography for the transverse-field Ising chain, which has a natural scaling limit for the number of visible units M (i.e., the number of spins), we demonstrate almost perfect data collapse of the learning characteristics (Δ_θ vs. τ_θ) for different M in the independent- and correlation-learning regimes upon rescaling by the total correlation $C_{\text{tot}}(p)$ of the target distribution (see Fig. 2c in particular). This indicates that a similar tradeoff is to be expected in larger systems as well.

- We included results for RBMs trained on the MNIST dataset as a standard testbed for application-like problem types and sizes. As observed by the reviewer, too, the exact loss Δ_θ is no longer available (not least because the target distribution $p(x)$ is not known). Nevertheless, the approximate loss measures introduced in the revised manuscript indicate similar behavior in the MNIST example as in our other examples where Δ_θ can be evaluated exactly.

- what is the impact of the parameter initialization procedure on the regimes being presented?

We have moved the information about the initial conditions from the supplement to the main text (see beginning of “Mechanism”) and expanded the discussion to address the above questions. We also provide examples for other initializations, including ones with significant correlations, in the new Supplementary Note S3.

- what is the impact of the learning rate and other details of the learning algorithm (the paper does include comparison of the mini batch size and order of CD)?

Data for different learning rates had already been part of Fig. 3 (see different rows) and Fig. 4 (now Fig. S7, see bottom-left panel); see also lines 273–282. With the learning rate η , the batch size B , the order of contrastive divergence n_{CD} , and the number of training samples $|S|$, we believe that we discuss the most important hyperparameters of RBM training. In addition, we include results for the persistent CD algorithm in the revised manuscript (Figs. 2b, 4, S7a) as a variation of the approximation scheme of the model averages in Eq. (4).

- can the behaviour of τ be compared with the sampling complexity and their relation verified / evaluated experimentally?

We assume that “sampling complexity” here refers to the minimum number of training samples required to reach $\Delta_\theta \leq \varepsilon$ with probability $1 - \delta$ for some preset $\delta, \varepsilon > 0$. Formally, this sampling complexity is infinite in the setup of our manuscript (for sufficiently small δ, ε). The reason is that the standard RBM training algorithms based on Markov-chain sampling (e.g., contrastive divergence) are inconsistent (the updates are biased), i.e., they will typically not converge to an optimal representation even if the number of training samples $|S|$ is infinite (see, for example, Fig. 3e).

However, we believe that this issue relates to the question for which $|S|$ the onset of the degradation regime is purely caused by limited expressivity of the RBM (N too small) or biased gradients (n_{CD} too small), but not by overfitting ($|S|$ too small). This question is investigated in some detail in the manuscript (see lines 314–371 in particular), and we tried to vary the relevant hyperparameters to showcase those different mechanisms behind degradation (see Figs. 2 and 3e in particular).

- can one obtain a convergence rate result for the training error / generalization error, or describe the training error / generalization error against the training data / computation time / computation effort?

We are not sure what a “convergence rate result for the training error/generalization error” would be. As observed above, we generally do not expect the loss to converge because of the biased training algorithms. (For example, even the training error typically deteriorates in later training stages as illustrated by the solid lines in Fig. 2b.)

Relating the evolution of the loss to the real computation time/effort is indeed a very relevant problem for applications. Unfortunately, it appears rather difficult to us to make general statements because it is influenced by a number of external circumstances (e.g., number and speed of CPUs and/or GPUs, memory access, software implementation, ...). In fact, we believe that the mutual

dependence of Δ_θ and τ_θ which we map out here is among the simplest general indicators for the relationship between loss and computational effort. If all external influences are fixed, the computational effort is essentially proportional to τ_θ since the latter quantifies how many Markov-chain samples need to be generated to reach the desired evaluation accuracy of the model properties.

- can one obtain a more explicit description of the qualitative differences in the Δ_θ - τ_θ curves depending on the type of target distribution / size of the RBM?

We admit that we do not have a full understanding of these issues yet and see it as an important direction for future research (see also end of “Discussion”). Some thoughts on characteristics of the target distribution that facilitate or impede learning by RBMs are provided in the Supplementary Notes (Secs. S5 C and S5 D in particular).

Other comments

- * line 138 ‘‘we would then need on the order of τ_θ R correlated Markov-chain samples [33]’’ Explain why and indicate specific statement in the referenced work.

We have substantially extended our description and discussion of integrated autocorrelation times in general and our sampling-efficiency measure τ_θ in particular in the sections “Accuracy and efficiency” and “Methods.” Specifically, we reproduce below Eq. (17) the argument from Ref. [33] (see Sec. 2 therein) about why τ_θ quantifies how many more samples are needed from the Markov chain compared with independent ones.

- * Section Mechanism. Here a discussion of the initialisation is needed.

As mentioned above, the discussion of the initialization scheme has now been included in the main text in the beginning of “Mechanism.”

- * In line 202, add explanation for why the decomposition $D(p||p^\wedge)=C(p)+D(p||p^\wedge)$ is valid. This being claimed to then conclude the weaker statement $\Delta \geq C_{\text{tot}}(p)$.

We now provide the derivation in Supplementary Note S5 A.

- * line 214 ‘‘no significant correlations between the RBM’s visible units build up as long as $\tau_\theta \approx 1$ ’’
Is this not true by definition?

Since τ_θ is based on the autocorrelation times of a specific set of observables (namely, the visible units x_i), it could possibly miss or vastly underestimate correlations that could be exposed by other observables. We are fairly confident that this is not the case, as we explain in the new segment in “Methods” on autocorrelation times and verify by simulations now included in the Supplementary Information (Sec. S4). The observation in the cited paragraph that τ_θ and $C_{\text{tot}}(\hat{p}_\theta)$ go hand in hand is another piece of evidence that τ_θ captures the relevant correlations. We hope that the extended discussion of autocorrelation times in our revised manuscript clarifies why this observation is nontrivial.

- * line 222 ‘‘whose standard deviation’’ This is defined later but would be good to define it by the time it appears first in the next.

We have moved the corresponding paragraph to the Supplementary Information (Sec. S5 C) and added the definition of σ_w .

- * line 243 ‘‘ σ_x ’’ add definition here

Since the entire model is defined more comprehensively in the figure caption, we have decided to include the definition of σ^x and σ^z there (see also next item). We refer to this figure caption in the cited paragraph of the main text now for clarification.

* Fig 2. definition of σ^z and σ^x basis missing. ‘‘(b) ... points’’ points are not visible in the plot (this could be more visible if reducing the size). ‘‘solid lines’’ these are also not visible. ‘‘All data points correspond to averages over 5 independent training runs’’ This could be explained more clearly.

The meanings of σ^z and σ^x are now explained in the figure caption. Regarding the clarity of the plots, which was also criticized by Reviewer 1, we have slightly decreased the size of the data points, but remain somewhat reluctant to reduce the size further because it would sacrifice information encoded in the colors. Instead, we have decided to add an explanatory remark regarding the visualization in the main text (see lines 193–198); see also reply to Reviewer 1.

As explained in the figure caption, the solid lines are sometimes hidden underneath the data points. This means that $\tilde{\Delta}_\theta^{(S)}$ coincides with Δ_θ , but it does not provide any further insights. Since the main quantity of interest is Δ_θ (i.e., the data points), we have decided to render the points above the lines.

The averaging over different runs in Figs. 2 and 3 is now explained in the beginning of the section ‘‘Mechanism.’’ Note that there is no such averaging performed in the new Fig. 4 because the goal of this figure is to inspect the situation typically encountered in applications.

* Fig 3. is also difficult to read.

Similarly as for Fig. 2 (see reply to the previous comment), we adjusted the sizes of the data points.

* line 336 ‘‘when the accuracy of the model approaches that of the empirical distribution’’ This probably needs rephrasing, since it makes no sense as stated.

We have reformulated the criticized passage.

* Fig 4. ‘‘50000 samples’’ Here it would be good to explain more clearly what is the variability of the samples, since most pixels are occupied by a small number of different patterns.

We are not entirely sure what information is being referred to by the ‘‘variability of the samples.’’ We have added information about the total number of 5×7 images showing one of the ten patterns in the figure caption, which is 40 507 353. In other words, this is the total number of images x that have a nonvanishing probability $p(x)$ in the target distribution. However, some of the images are (much) more likely to be generated than others. A detailed description and characterization of the distribution is provided in the Supplementary Information (Sec. S2 C), including the probability distribution, its entropy and total correlation, and the frequencies of the respective patterns. If more of that information should be stated in the main text, we would be thankful for a hint at which parts to include.

* line 454 ‘‘what distributions can be represented efficiently by RBMs... symmetries that make it possible to represent the correlations’’ Here two references appear useful to mention, which explain how correlations among visible units are represented in RBMs and the types of interactions that can be represented by hidden units connected with visible units:
 - Younes. Synchronous Boltzmann machines can be universal approximators. Applied Mathematics Letters, 9(3):109–113, 1996.
 - Montufar and Rauh. Hierarchical Models as Marginals of Hierarchical Models. International Journal of Approximate Reasoning ; DOI: 10.1016/j.ijar.2016.09.003; Date: September, 2016.

We thank the reviewer for pointing us to these references. We have included them in the cited paragraph as well as in one other place (line 308) referring to the universal approximation property of RBMs.

* line 487 ‘‘decreases towards zero as $|\theta_k|$ increases’’. Although this tends to happen, the statement is not exactly correct. It is possible to have all individual parameters have parameters with magnitude going to infinity without the entropy going to zero.

The criticized statement was indeed inaccurate. We have adapted the wording along the lines suggested by the reviewer. The revised Supplementary Information also contains a more detailed analysis of the scaling relation between the magnitude of the weights and the autocorrelation time (Sec. S5 C and Fig. S8).

* line 578 ‘‘Data availability... upon reasonable request’’ I would suggest that a supplementary material with the computer code to replicate the experiments be submitted with the paper.

We will make the computer code available in a public repository. For review, please refer to the file `source_code.zip` submitted with this revision. It includes a readme file with instructions how to compile and run the program as well as examples for the different experiments. We added a ‘‘Code availability’’ statement to the manuscript.

* Another article that appears relevant in the context of this work is:
Shun-ichi Amari, Information Geometry on Hierarchy of Probability Distributions, IEEE Transactions on Information Theory 47(5):1701 - 1711, August 2001.

Drawing a direct connection to information geometry would indeed be a highly interesting extension. A particularly intriguing question to us is whether there exists a hierarchy of correlations within the correlation-learning regime, similar to the pronounced transition from the independent-learning to the correlation-learning regime. Most notably, the $g = 1$ example from Fig. 2d, which is also discussed in more detail in Supplementary Note S5 D, led us to speculate about such a hierarchy. Unfortunately, the analysis we have conducted so far, which includes correlations/marginal distributions up to the fourth order, did not yield any definite evidence of such a hierarchy. Notwithstanding, we mention the idea and the suggested reference (along with a couple of other ones) towards the end of the ‘‘Discussion’’ in the revised manuscript (lines 596–602).

REVIEWERS' COMMENTS

Reviewer #1 (Remarks to the Author):

The authors have addressed the comments by both reviewers in great detail. They provided all missing information and made the manuscript very clear. The additional section "Towards applications" promotes the validity of the work beyond the theoretical studies provided.

The authors have also modified the figures, which are now much better to understand. As a small comment I find that all the plots are very small, especially when considering the amount of information they carry. The authors might consider increasing the figure sizes as a whole.

Apart from this minor point I think that the manuscript is now very complete and clearly written. It provides a significant contribution to the research community and I suggest acceptance for publication.

Reviewer #2 (Remarks to the Author):

The revision addressed several points from my previous review, particularly by 1) adding experiments with larger models and different settings, 2) expanding discussions and formulating a few possible recommendations for practical applications. Some aspects are left to be developed in more detail in the future, such as an adaptive CD. Parts of the document have also been reorganised for clarity, particularly the discussion of initial conditions. Importantly, computer code is being added to the submission. On the other hand, a deeper theoretical analysis remained unaddressed. As mentioned in my previous review, however, I do not think this is strictly necessary if the experimental part and discussion are sufficiently strong. The revision has improved these aspects.

Overall I find the manuscript makes interesting observations with sensible motivations and reasonable experiments. I find the article fair in terms of "represent[ing] important advances of significance to specialists". I think it is worthy of publication and suitable for a multidisciplinary journal.

Minor items:

The manuscript indicates that "findings are based on analytical arguments and numerical experiments" and "we quantitatively map out the tradeoff relationship between accuracy and efficiency". I found this confusing, because while indeed a formula is stated and it is found to be reasonably consistent with certain experiments, no mathematical analysis or analytical arguments are presented to substantiate the hypothesised relation.

For instance, for the independent-learning regime there is a pointer to Supplement S5, which however only gives a decomposition of the KL but does not show that learning will actually proceed in this way. Then a note is made that "Quantitatively, we find that the functional dependence ... " but only a pointer to figures containing experimental data and no mathematical analysis is given.

After eq 6 it would be good to explain x_i (which is written next to $x_i^{(0)}$ and $x_i^{(n)}$). Then the independence from the training algorithm is under the assumption that there is no training between $x^{(0)}$ and $x^{(n)}$, correct?, this could also be pointed out.

In line 141 it will be good to point at a specific result from reference [33], particularly since it is a book.

In line 175 the initialization procedure should be compared with schemes proposed in RBM practice, eg from Hinton's practical guide and other.

In Fig 4. "Gray pixels should either be dark or lie outside the image boundaries." This is not clear.

Reply to Reviewer Reports (2nd Round)

We are grateful to both reviewers for assessing our work once more and for recommending its publication.

Below, we reply to their comments in detail. All text changes in the manuscript have been color-coded.

Reviewer 1

The authors have addressed the comments by both reviewers in great detail. They provided all missing information and made the manuscript very clear. The additional section "Towards applications" promotes the validity of the work beyond the theoretical studies provided.

The authors have also modified the figures, which are now much better to understand. As a small comment I find that all the plots are very small, especially when considering the amount of information they carry. The authors might consider increasing the figure sizes as a whole.

We have increased the overall size of Figs. 2–4 and also adjusted the layout of Figs. 3 and 4 to increase the display area for the main panels.

Apart from this minor point I think that the manuscript is now very complete and clearly written. It provides a significant contribution to the research community and I suggest acceptance for publication.

We thank the reviewer for his/her positive evaluation.

Reviewer 2

The revision addressed several points from my previous review, particularly by 1) adding experiments with larger models and different settings, 2) expanding discussions and formulating a few possible recommendations for practical applications. Some aspects are left to be developed in more detail in the future, such as an adaptive CD. Parts of the document have also been reorganised for clarity, particularly the discussion of initial conditions. Importantly, computer code is being added to the submission. On the other hand, a deeper theoretical analysis remained unaddressed. As mentioned in my previous review, however, I do not think this is strictly necessary if the experimental part and discussion are sufficiently strong. The revision has improved these aspects.

Overall I find the manuscript makes interesting observations with sensible motivations and reasonable experiments. I find the article fair in terms of ‘represent[ing] important advances of significance to specialists’. I think it is worthy of publication and suitable for a multidisciplinary journal.

We thank the reviewer for his/her positive evaluation.

Minor items:

The manuscript indicates that ‘findings are based on analytical arguments and numerical experiments’ and ‘we quantitatively map out the tradeoff relationship between accuracy and efficiency’. I found this confusing, because while indeed a formula is stated and it is found to be reasonably consistent with certain experiments, no mathematical analysis or analytical arguments are presented to substantiate the hypothesised relation.

For instance, for the independent-learning regime there is a pointer to Supplement S5, which however only gives a decomposition of the KL but does not show that learning will actually proceed in this way. Then a note is made that ‘‘Quantitatively, we find that the functional dependence ...’’ but only a pointer to figures containing experimental data and no mathematical analysis is given.

We changed the wording of the criticized passages in the abstract and in line 57 as well as in line 169 to avoid raising false expectations of a mathematically rigorous derivation.

After eq 6 it would be good to explain x_i (which is written next to $x_i^{(0)}$ and $x_i^{(n)}$). Then the independence from the training algorithm is under the assumption that there is no training between $x^{(0)}$ and $x^{(n)}$, correct?, this could also be pointed out.

Since x_i here refers to a sample from the model distribution $\hat{p}_\theta(x)$ and $x_i^{(0)}$ is such a sample, we replaced x_i by $x_i^{(0)}$ for uniformity of notation in the sentence below Eq. (6) and also in Eq. (S17).

By ‘‘independence from the training algorithm,’’ we mean that the Markov chains to estimate τ_θ are initialized in the stationary distribution as specified in the paragraph below Eq. (6), whereas different training algorithms such as CD or PCD use different initialization schemes when calculating gradient estimates (see paragraph above Eq. (6)). We added a sentence below Eq. (6) to clarify this.

In line 141 it will be good to point at a specific result from reference [33], particularly since it is a book.

We explicitly point to Sec. 2 of the specified reference now.

In line 175 the initialization procedure should be compared with schemes proposed in RBM practice, eg from Hinton’s practical guide and other.

We added a corresponding remark in lines 188–189 and 249–253. We also point to Supplementary Note 3 again, where other initialization schemes are discussed in more detail.

In Fig 4. ‘‘Gray pixels should either be dark or lie outside the image boundaries.’’ This is not clear.

We reformulated the sentence and added a pointer to the provided examples for illustration.